# Privileged Sensing Scaffolds Reinforcement Learning

**Edward S. Hu**[1]     **James Springer**[1]     **Oleh Rybkin**[2]     **Dinesh Jayaraman**[1]
[1]University of Pennsylvania     [2]UC Berkeley
hued@seas.upenn.edu

## Abstract

We need to look at our shoelaces as we first learn to tie them but having mastered this skill, we can do it from touch alone. We call this phenomenon "sensory scaffolding": observation streams that are not needed by a master might yet aid a novice learner. We consider such sensory scaffolding setups for training artificial agents. For example, a robot arm may need to be deployed with just a low-cost, robust, general-purpose camera; yet its performance may improve by having privileged training-time-only access to informative albeit expensive and unwieldy motion capture rigs or fragile tactile sensors. For these settings, we propose *Scaffolder*, a reinforcement learning approach which effectively exploits privileged sensing in critics, world models, reward estimators, and other such auxiliary components that are only used at training time, to improve the target policy. For evaluating sensory scaffolding agents, we design a new "S3" suite of ten diverse simulated robotic tasks that explore a wide range of practical sensor setups. Agents must use privileged camera sensing to train blind hurdlers, privileged active visual perception to help robot arms overcome visual occlusions, privileged touch sensors to train robot hands, and more. *Scaffolder* easily outperforms relevant prior baselines and frequently performs comparably even to policies that have test-time access to the privileged sensors. Website: https://penn-pal-lab.github.io/scaffolder/

## 1 Introduction

It is well-known that Beethoven composed symphonies long after he had fully lost his hearing. Such feats are commonly held to be evidence of mastery: for example, novice typists need to look at the keyboard to locate keys but with practice, can graduate to typing without looking. Thus, sensing requirements may be different *during* learning versus *after* learning. We refer to this as "sensory scaffolding", drawing inspiration from the concept of scaffolding teaching mechanisms in psychology that provide temporary support for a student (Wood et al., 1976; Vygotsky et al., 2011), like training wheels when learning to ride a bicycle.

For artificial learning agents such as robots, sensory scaffolding permits decoupling the observation streams required at test time from those that are used to train the agent. The sensors available in a deployed robot are often decided by practical considerations such as cost, robustness, size, compute requirements, and ease of instrumentation, e.g., autonomous cars with only cheap and robust RGB camera sensors. However, those considerations might carry less weight at training time, so a robot learning practitioner may choose to scaffold policy learning with *privileged information* (Vapnik & Vashist, 2009) from extra sensors available only at training. In the case of the cars above, the manufacturer might equip a small fleet of training cars with expensive privileged sensors like lidar to improve RGB-only driving policies for customers to install in their cars.

What learning mechanisms might permit artificial agents to improve by exploiting such privileged, training-time sensors? Considering reinforcement learning (RL) algorithms, we observe that while their primary output is usually a policy, they often employ an elaborate training apparatus with value functions, representation learning objectives, reward estimators, world models, and data collection policies. While the output policy must only access pre-determined target sensors, our key insight is that *this training apparatus offers natural routes for privileged observation streams to influence policy learning.*

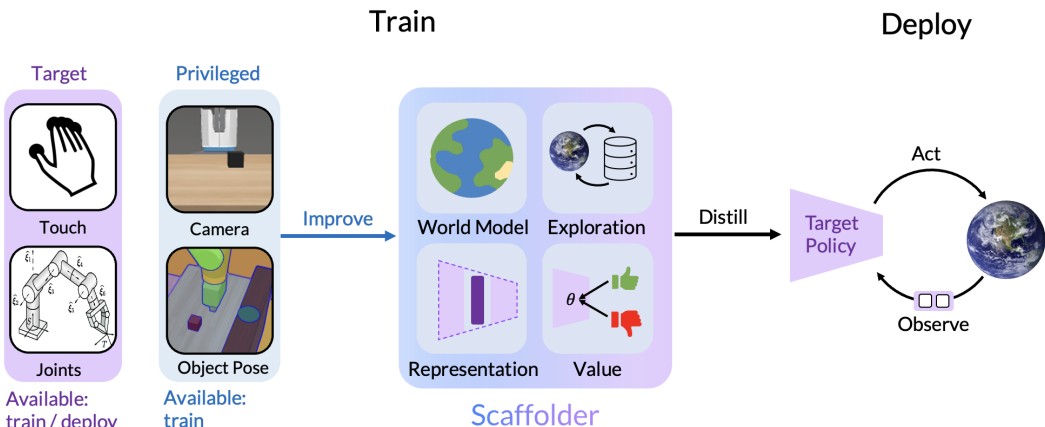

Figure 1: Learning a policy to operate from partial observations can be aided by access to privileged sensors exclusively during training. *Scaffolder* improves world models, critics, exploration, and representation learning objectives to synthesize improved target policies.

This insight directly motivates *Scaffolder*, a novel model-based reinforcement learning (MBRL) approach that "scaffolds" each training component of RL by providing it access to privileged sensory information. Figure 1 shows a schematic. MBRL algorithms learn an environment simulator, or world model, from experience, and then train policies on synthetic experiences collected within this simulator, potentially mediated by value functions and reward estimators. In *Scaffolder*, rather than training a low-fidelity world model on the impoverished target observations, we train a "scaffolded world model" with privileged observations that more accurately models environment dynamics. This enables more realistic training experience synthesis, and better credit assignment using scaffolded value functions and reward estimators. Further, exploratory data collection, both in the real environment and within the world model, can now be better performed by employing a scaffolded exploration policy. Finally, the scaffolded observations also enable learning improved representations of the test-time target sensor observations.

Our key contributions are: **(1)** We study policy learning with privileged information in a novel and practically well-motivated "sensory scaffolding" setting, where extra sensors are available to an agent such as a robot at training time. **(2)** We propose *Scaffolder*, a MBRL method that extensively utilizes privileged observations to scaffold each auxiliary component of RL. **(3)** We validate *Scaffolder* extensively against prior state of the art on a new Sensory Scaffolding Suite (S3). S3 contains ten diverse environments and sensor setups, including privileged active visual perception for occluded manipulation policies, privileged touch and pose sensors for dexterous manipulation policies, and privileged audio and sheet music for training "blind and deaf" piano-playing robots. See Figure 2. **(4)** Through detailed analyses and ablation studies, we study the relative impacts of scaffolded learning mechanisms through which privileged sensing impacts policy learning. We find all components are important and that each component's contribution depends on task-specific properties. Finally, we show empirically that *Scaffolder* improves the agent's estimates of the true RL objective function, thus providing improved learning signals to drive policy improvement.

## 2 PROBLEM SETUP, NOTATION, & PRIOR WORK

**The Sensory Scaffolding Problem Setting:** Consider the setup in Figure 1: a robot policy must use only target observations $o_t^- = \{$ proprioception, touch $\}$ to pick up a block, whose pose is unknown. It is common to model such problems as partially observable Markov decision process (POMDP) (Kaelbling et al., 1998) and train the policy $\pi^-(a_t \mid o_t^-)$ to select appropriate robot actions $a_t \in \mathcal{A}$. However, note in Figure 1 that the training process has access to additional *privileged* camera observations $o_t^p$: thus, the policy $\pi^-(a_t \mid o_t^-)$ effectively trains in a different, "scaffolded" POMDP with observations $o_t^+ = [o_t^-, o_t^p]$. These scaffolded observations $o_t^+$ may still only be partial in that they don't reveal the full environment state. Indeed they often are partial in our evaluation settings, but they nevertheless contain strictly more information about the environment state $s_t \in \mathcal{S}$ than the

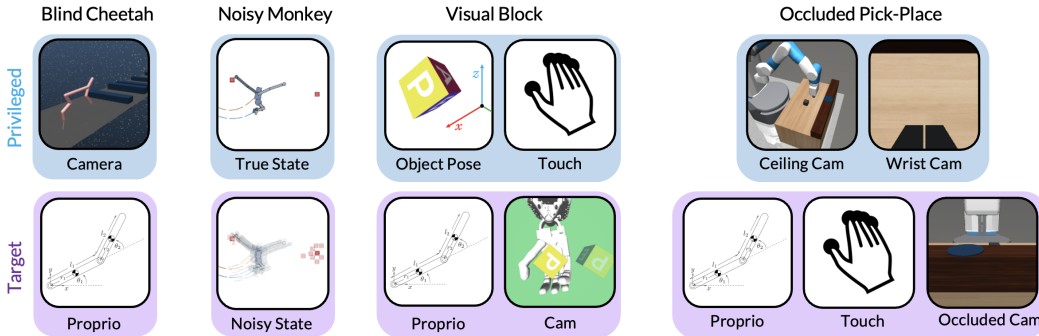

Figure 2: Sensory Scaffolding Suite (S3). We visualize four out of our ten diverse tasks, each exploring different restricted sensing scenarios such as proprioceptive-only inputs, noisy sensors, images, and occluded or moving viewpoints. We evaluate the enhancement of policy training using privileged sensors like multiple cameras, controllable cameras, object pose, and touch sensors. Refer to Appendix E.4 for details on all environments.

target observations $o_t^-$. In the picking task, even though the policy will operate blind at test time, the learning procedure can use noisy knowledge of the block pose, as revealed through the privileged camera observations.

**Prior Work:** We now review several prior lines of work on reinforcement learning (RL) with privileged information which may apply to this sensory scaffolding problem. See Appendix A for extended discussion. Perhaps the most straightforward vehicle for privileged information in RL is reward. Analogous to labels in supervised learning, RL rewards specify the task, are only present at training time, and may therefore exploit privileged sensors. For example, Schenck & Fox (2017) instrument privileged thermal cameras to gauge fluid levels to train an image-based pouring policy. In another interesting example, Huang et al. (2022a) exploit privileged *interactive* sensing behaviors for reward estimation, such as pulling at a door to evaluate whether a door locking task is properly completed. Further, nearly all sensorimotor RL in simulation uses privileged low-dimensional simulator state $o_t^+ = s_t$ to inform task rewards and thus may be seen as implementing sensory scaffolding in a limited way. More pertinent to us are methods that exploit privileged observations in other training-specific components of RL beyond just the reward. We discuss these below.

- **Privileged Critics:** Actor-critic methods commonly condition the critic on privileged information (i.e. $v(o^+)$), usually low-dimensional simulator state, while training the actor policy $\pi^-(a \mid o^-)$ on target observations like high dimensional images (Pinto et al., 2018; Andrychowicz et al., 2018). These methods assume that target observations contain full information about the state, and struggle when this assumption is violated (Baisero & Amato, 2021).
- **Privileged Policies:** These methods train a privileged teacher $\pi^+$ to guide the student target policy $\pi^-$. A common failure mode here is the "imitation gap"(Weihs et al., 2021; Swamy et al., 2022): the student cannot recover the teacher's actions given impoverished inputs $o^-$. This is typically mitigated by incorporating an additional reinforcement loss into the student's learning objective (Rajeswaran et al., 2017; Weihs et al., 2021; Nguyen et al., 2022; Shenfeld et al., 2023). Aside from direct imitation (Chen et al., 2020), privileged teachers can also benefit the student's exploration by sharing data (Schwab et al., 2019; Shenfeld et al., 2023; Kamienny et al., 2020; Weigand et al., 2021) or defining auxiliary rewards for the student (Walsman et al., 2022).
- **Privileged World Models:** Most privileged sensing strategies employ model-free RL methods, with few exploring enhancements to model-based RL using privileged sensors. Seo et al. (2023) improves DreamerV2 (Hafner et al., 2022) by training the single-view policy representation on multi-view data. Recently, Informed Dreamer (Lambrechts et al., 2023) improve DreamerV3's (Hafner et al., 2023) representation and world modelling via privileged information prediction.
- **Privileged Representation Learning Objectives:** Privileged observations are commonly used to train representations for high-dimensional, image-based tasks, for example, by leveraging privileged sensors like additional views (Sermanet et al., 2018; Seo et al., 2023) or segmentations (Salter et al., 2021). Several recent works have leveraged privileged simulator state for sim2real applications (Lee et al., 2020; Kumar et al., 2021; Qi et al., 2023), training a policy conditioned on

target observations and predicted simulator states. These methods have demonstrated impressive results in quadruped locomotion and dexterous manipulation.

Summarizing, nearly all prior works focus on one route for privileged observations to influence target policy learning, and many suffer from assumptions about the privileged observation $o^p$ or target observation $o^-$ that do not hold in all settings. *Scaffolder* makes no such assumptions and exploits the scaffolded observations $o^+ = [o^p, o^-]$ through multiple routes integrated into one cohesive RL algorithm. We also compare our method empirically against representative methods from each category of prior work, demonstrating significant and consistent gains on a large suite of tasks.

## 3  *Scaffolder*: IMPROVING POLICY TRAINING WITH PRIVILEGED SENSORS

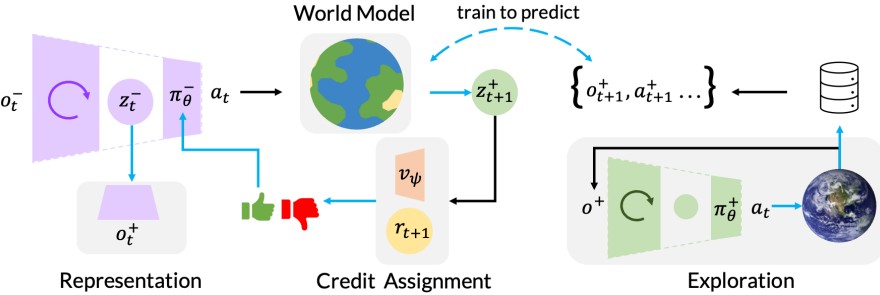

Figure 3: *Scaffolder* uses scaffolded observations to improve all components of training: world modelling, credit assignment, exploration, and policy representation.

We now describe *Scaffolder*, a model-based reinforcement learning (MBRL) approach that uses scaffolded observations to comprehensively improve training mechanisms. MBRL algorithms learn an environment simulator, or world model (WM), from data and then train policies inside the simulator, facilitated by learned reward estimators and value functions. In the sensory scaffolding setting, we suggest a conceptually simple improvement: train the WM on scaffolded observations $o^+$ instead of impoverished target observations $o^-$. The scaffolded WM is then used to improve target policy training, credit assignment, and exploration. We also improve the target policy representation learning objective with $o^+$. See Figure 3 for a visualization.

**Scaffolded World Model.**  We build on DreamerV3 (Hafner et al., 2023), a MBRL method well-known for its generality (see Appendix C.1 for DreamerV3 details). DreamerV3's WM is implemented as a Recurrent State-Space Model (Hafner et al., 2020). The world model serves two purposes: it acts as an environment simulator to train the policy and serves as the policy's recurrent state encoder for aggregating observation history. *Scaffolder* learns an additional "scaffolded" WM trained on $o^+$ to replace the target WM (trained on $o^-$) for policy training. We still retain and train the original target world model so we can encode states for target policy execution. We define both world models below.

| | Target | Scaffolded | | Target | Scaffolded |
|---|---|---|---|---|---|
| Embedding: | $e_t^- = \text{emb}_\phi^-(o_t^-)$ | $e_t^+ = \text{emb}_\phi^+(o_t^+)$ | Reward: | $p_\phi^-(r_t \mid z_t^-)$ | $p_\phi^+(r_t \mid z_t^+)$ |
| Dynamics: | $p_\phi^-(z_t^- \mid z_{t-1}^-, a_{t-1})$ | $p_\phi^+(z_t^+ \mid z_{t-1}^+, a_{t-1})$ | Decoder: | $p_\phi^-(o_t^- \mid z_t^-)$ | $p_\phi^+(o_t^+ \mid z_t^+)$ |
| Posterior: | $q_\phi^-(z_t^- \mid z_{t-1}^-, a_{t-1}, e_t^-)$ | $q_\phi^+(z_t^+ \mid z_{t-1}^+, a_{t-1}, e_t^+)$ | Continue: | $p_\phi^-(c_t \mid z_t^-)$ | $p_\phi^+(c_t \mid z_t^+)$ |
| Emb. Predictor: | | $p_\phi^+(e_t^- \mid z_t^+)$ | | | |

The embedding is a low-dimensional representation of the observation. The recurrent posterior acts as the state encoder and infers the current state given history and current observation, while the dynamics predicts the current state given only history. The decoder, reward, and continue heads reconstruct the observations, rewards, and termination signals from the trajectory data. Both WMs are trained to encode and predict collected trajectories using DreamerV3's world model learning objectives (see Appendix C for full equations).

**Scaffolded Critic and Reward.**  DreamerV3, without privileged sensing, optimizes the target policy $\pi_\theta^-(a \mid z^-)$ by maximizing the policy's estimated return, which is computed by evaluating

the learned reward and critic networks over synthetic trajectories generated by the world model. *Scaffolder* improves the return estimate by using more accurate synthetic experience from the scaffolded WM, and better reward and value estimates from the scaffolded reward $p_\phi^+(r \mid z^+)$ and critic $v_\psi^+(z^-, z^+)$. Note that the critic requires both $z^-, z^+$ to remain unbiased, see Appendix C.2 for details. To evaluate the policy, the scaffolded critic and reward networks require trajectories with scaffolded latent states. We describe how to acquire such trajectories in imagination.

▶ **Nested Latent Imagination (NLI)**: NLI is a procedure to roll out $\pi_\theta^-$ inside the scaffolded world model to generate synthetic training data. See Figure 4 for a visualization. At time $t$, the target policy selects action $a_t = \pi_\theta^-(z_t^-)$ to execute inside the scaffolded WM, generating next scaffolded latent $z_{t+1}^+$ and reward $r_{t+1}^+$. To continue the rollout by querying the policy at time $t+1$, the scaffolded $z_{t+1}^+$ must be *translated into its corresponding target latent $z_{t+1}^-$*. This is

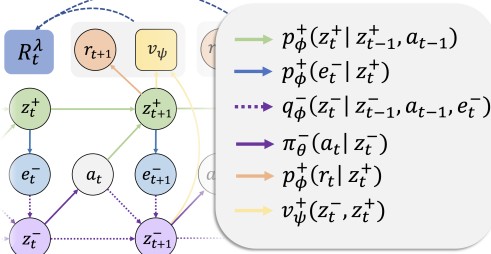

Figure 4: Nested Latent Imagination.

achieved by first employing a new "target embedding prediction" model $p_\phi^+(\hat{e}_{t+1}^- \mid z_{t+1}^+)$, trained so that $\hat{e}_{t+1}^- \approx \text{emb}_\phi^-(o_{t+1}^-)$. Then, $\hat{e}_{t+1}^- \sim p_\phi^+(\hat{e}_{t+1}^- \mid z_{t+1}^+)$, fed into the target posterior $q_\phi^-(z_{t+1}^- \mid z_t^-, a_t, \hat{e}_{t+1}^-)$ yields the desired translated target latent $z_{t+1}^-$, compatible with the target policy $\pi_\theta^-$. This completes one step of the trajectory generation and is repeated $H$ times to create a trajectory $\tau = [z_1^+, z_1^-, a_1, \dots z_H^+, z_H^-]$.

▶ **Computing the TD($\lambda$) return**: Now, we show how to improve target policy training with scaffolded observations and components. All RL methods aim to maximize the true discounted policy return. Policy updates in Dreamer follow the gradient of the TD($\lambda$) return, an estimate of the true return that is mediated by the world model's latent state, reward predictions and value function. We modify DreamerV3's policy objective to use scaffolded dynamics, scaffolded reward estimator, and scaffolded critic instead of the impoverished target components. We highlight the terms that depend on the scaffolded components in blue. The target policy objective is defined as

$$\mathcal{J}(\theta) = \sum_{t=1}^{T} \mathbb{E}_{\pi_\theta^-, p_\phi^+}\Big[ R_t^\lambda + \eta \mathrm{H}\big[\pi_\theta^-(a_t \mid z_t^-)\big]\Big] \tag{1}$$

where the first term maximizes the policy's TD($\lambda$) return, the second term is an entropy regularizer, and the expectation is over the imagined trajectories generated using the above NLI procedure. The TD($\lambda$) return is the weighted average of $n$-step returns, defined as:

$$R_t^\lambda \doteq r_t + \gamma c_t\big((1 - \lambda)v_\psi^+(z_{t+1}^-, z_{t+1}^+) + \lambda R_{t+1}^\lambda\big) \qquad R_T^\lambda \doteq v_\psi^+(z_T^-, z_T^+) \tag{2}$$

This modified objective now benefits from the scaffolded components in the following ways. First, the imagined trajectories are generated with the scaffolded dynamics, which can better approximate the true dynamics compared to the target dynamics. Next, the reward and value estimates can now use $z^+$, which contains potentially relevant information not available in $z^-$ to improve credit assignment. This all results in a more accurate TD($\lambda$) return estimate of the true return, which in turn better optimizes the policy. Thus, scaffolding the world model directly improves the RL training signals provided to the target policy $\pi_\theta^-$ at each update.

While only the WM and critic are directly involved in the policy update above, they are trained on *exploration* data, and the policy itself operates on a *representation $z^-$*. These can both be improved by using privileged observations, as we discuss below.

**Scaffolded Exploration Policy.** We train a scaffolded task policy $\pi_\theta^+(a_t \mid z_t^+)$ inside the scaffolded world model. While $\pi_\theta^+$ cannot directly run at test time since privileged observations would be missing, it can generate exploratory data to train $\pi_\theta^-$. The intuition is that the scaffolded $\pi_\theta^+$ learner becomes more performant more quickly, so it explores task-relevant states beyond the current reach of $\pi_\theta^-$. We use a 1:1 ratio of $\pi_\theta^+$ and $\pi_\theta^-$ rollouts for exploration. We choose not to enforce any imitation objective between $\pi^+$ and $\pi^-$ to avoid the "imitation" gap (Section 2).

**Scaffolded Representation Learning Objective.** Finally, following prior works (Pinto et al., 2018; Lambrechts et al., 2023) that have found it useful to regress privileged information for representation

learning, we train an auxiliary decoder from $z^-$ to the scaffolded observation $o^+$, to improve $z^-$. We expect this to be helpful when $o^p$ is recoverable from $o^-$, i.e. inferring object poses from images.

## 4 EXPERIMENTS

We aim to answer the following questions. **(1)** Does *Scaffolder* successfully exploit privileged sensing to improve target policy performance? **(2)** How does *Scaffolder* compare to prior work on RL with privileged information? **(3)** What are the most critical components through which privileged observations influence target policy learning, and how do task properties affect this?

### 4.1 THE SENSORY SCAFFOLDING SUITE (S3) OF TASKS

We propose Sensory Scaffolding Suite (S3), a suite of 10 robotics-based tasks to evaluate *Scaffolder*, baselines, and future approaches in the sensory scaffolding problem setting. See Figure 5. As motivated in Section 1, robotics is a well-suited domain for studying sensory scaffolding: practical considerations such as cost, robustness, size, compute requirements, and ease of instrumentation often incentivize operating with limited sensing on deployed robots. In addition to standard definitions for RL environments, S3 pre-defines privileged and target sensors. S3 tasks are more general and difficult than prior sensory scaffolding tasks in a variety of ways. S3 tasks have continuous, observation and action spaces with complex dynamics. Furthermore, the privileged sensors are potentially high-dimensional and noisy observations, rather than ground truth, low-dimensional simulator state. S3 defines performance scores for each task: success rate for pen and cube rotation, number of branches swung for Noisy Monkey, and returns computed from dense rewards for the other tasks.

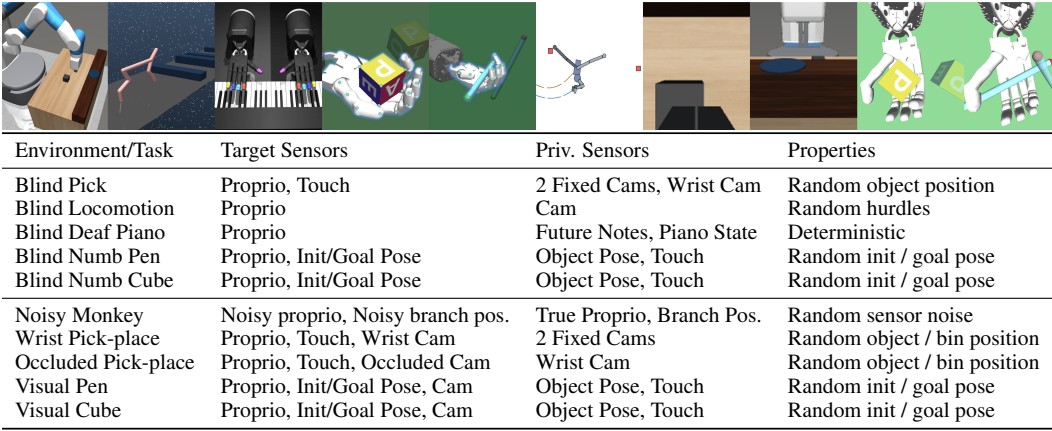

| Environment/Task | Target Sensors | Priv. Sensors | Properties |
|---|---|---|---|
| Blind Pick | Proprio, Touch | 2 Fixed Cams, Wrist Cam | Random object position |
| Blind Locomotion | Proprio | Cam | Random hurdles |
| Blind Deaf Piano | Proprio | Future Notes, Piano State | Deterministic |
| Blind Numb Pen | Proprio, Init/Goal Pose | Object Pose, Touch | Random init / goal pose |
| Blind Numb Cube | Proprio, Init/Goal Pose | Object Pose, Touch | Random init / goal pose |
| Noisy Monkey | Noisy proprio, Noisy branch pos. | True Proprio, Branch Pos. | Random sensor noise |
| Wrist Pick-place | Proprio, Touch, Wrist Cam | 2 Fixed Cams | Random object / bin position |
| Occluded Pick-place | Proprio, Touch, Occluded Cam | Wrist Cam | Random object / bin position |
| Visual Pen | Proprio, Init/Goal Pose, Cam | Object Pose, Touch | Random init / goal pose |
| Visual Cube | Proprio, Init/Goal Pose, Cam | Object Pose, Touch | Random init / goal pose |

Figure 5: The Sensory Scaffolding Suite (S3) of tasks. See Appendix E.4 for details.

We provide a high-level overview here; for more details, see Appendix E.4. Figure 5 provides key information about each task. **Tasks 1-5** in S3 focus on taking away privileged vision, audio, and touch to train blind, deaf, and numb robot policies that operate mainly from proprioception at test time. The robots span manipulators, legged robots, and dexterous hands, and the tasks include object picking, hurdling, pen rotation, and piano playing.

**Tasks 6-10** provide more target sensing to policies, particularly vision (either as raw RGB images, or as object recognition system outputs). Privileged senses are even more informative, such as multi-camera images, and "active" cameras that move with the robot's wrist. Equipped with the target sensors, monkey robots must swing between tree branches, dextrous hands must rotate objects, and 2-fingered robot arms must put objects away amidst occlusion on cluttered tables.

**Baselines.** In Section 2, we classified privileged RL methodologies based on the component they enhance with privileged information, such as the critic, policy, world model, and representation learning objective; we select representative baselines from each category to compare to *Scaffolder*. Wherever possible, we implement baselines over DreamerV3 (Hafner et al., 2023), the base MBRL agent for our method. We overview the baselines below, see Appendix E.3 for more details.

- **Unprivileged: DreamerV3.** We train DreamerV3 only on target observations. DreamerV3 is known for its consistency across benchmarks, thereby serving as a strong unprivileged baseline.

- **Critic: AAC.** For a privileged critic baseline, we use Asymmetric Actor Critic (AAC) (Pinto et al., 2018). AAC is model-free; we find that it is much more sample-inefficient, so we train it for 100M steps in each environment ($20 - 400\times$ more than *Scaffolder*).
- **Teacher Policy: DreamerV3+BC.** Here, we follow a common strategy from prior works (Weihs et al., 2021; Nguyen et al., 2022; Shenfeld et al., 2023) to train a privileged teacher policy and incorporate a teacher-imitating BC loss alongside RL rewards when training the target policy. We implement this in DreamerV3 and call it DreamerV3+BC. The weights for BC and RL objectives are selected through hyperparameter search for each environment.
- **World Model: Informed Dreamer.** Lambrechts et al. (2023) extends DreamerV3 to exploit privileged observations by incorporating a new objective term to decode privileged observations $o_t^+$ from target Dreamer state $z_t^-$.
- **Representation: RMA+.** Kumar et al. (2021) train policies with PPO using a privileged state representation analogous to $z^+$ and regress from target observations $o^-$ to $z^+$. This is liable to fail when $z^+$ may contain information that is not in $o^-$, but RMA has achieved impressive results for many robotics applications. To facilitate fast RMA training, we found it necessary to provide *ground truth simulator state* as privileged information; we call this RMA+. Like AAC, RMA+ is model-free, so we run it for 100M steps to permit it to learn meaningful policies.
- **Exploration Policy: Guided Obs.** Finally, Guided Observability (Weigand et al., 2021) collects better exploration data by dropping out the privileged information from policy inputs over time.

## 4.2 RESULTS

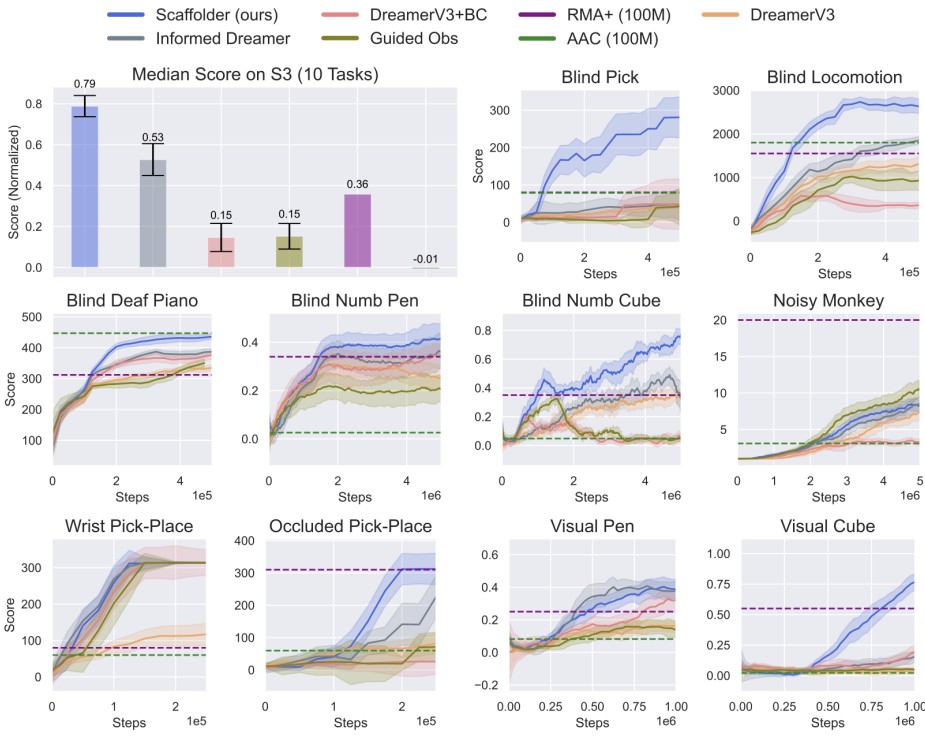

Figure 6: *Scaffolder* performs better than baselines across all tasks in learning speed and final performance, showing its generality. For each method, we report median score and standard error, and each method is run with 4-10 seeds aside from AAC (100M) and RMA+ (100M).

We evaluate the training performance and final task scores of each method. The normalized final median scores in the top left of Figure 6 are computed using the performance of the unprivileged DreamerV3 baseline as a lower bound (0.0) and the performance of a privileged DreamerV3 model that is trained and *evaluated* on $o^+$ as the upper bound (1.0). See Appendix E for more info.

Figure 6 demonstrates that *Scaffolder* achieves the highest aggregate median performance across all ten tasks. *Scaffolder* bridges $79\%$ of the gap between $o^-$ and $o^+$, just by having temporary access to $o^+$ at training time. In other words, much of the gap between the observations $o^-$ and $o^+$ might lie not in whether they support the same behaviors, but in whether they support *learning* them.

A closer look at the learning curves in Figure 6 reveals that *Scaffolder* learns more quickly in 8 out of 10 tasks. Even with its limited environment sample budget (between 250K to 5M), it outperforms or is competitive with AAC and RMA trained with 100M samples in 9 out of 10 tasks.

*Scaffolder* successfully exploits privileged sensors to learn performant policies despite severely limited target sensors - it plays an imperfect, but recognizable rendition of "Twinkle Twinkle Little Star", deftly rotates cubes to goal orientations without any object pose information, and actively moves a robot wrist camera to look for objects outside the field of view. See website for video examples of learned behaviors. Base DreamerV3 uniformly performs much worse.

The privileged baselines all exhibit a high degree of performance variance. They may excel in certain tasks yet are oftentimes worse than the non-privileged DreamerV3 baseline. Recall from Section 2 that these approaches focus on one route for privileged sensors to influence policy learning, and many make assumptions about the nature of privileged and target observations. This may be ill-suited to performing well on the diverse Sensory Scaffolding Suite. DreamerV3 operating on just the target sensors without any privileged info proves to be a surprisingly strong performer, even outperforming some prior approaches that access privileged information like AAC.

For example, RMA excels in Noisy Monkey, where regressing the privileged true state $o^p$ from noisy state $o^-$ is relatively easy, yet completely fails in Blind Pick. Indeed, we find that in environments like Blind Pick or Blind Locomotion with large information gaps between target and privileged observations, RMA, DreamerV3+BC, and Informed Dreamer tend to suffer. RMA fails because the target observations $o^-$ are not predictive of the privileged observations $o^p$ (e.g. proprioception is not usually predictive of object states). Informed Dreamer similarly fails because it also enforces the target world model to predict privileged observations from target observations. Finally, DreamerV3+BC fails in such cases due to the large differences in optimal behavior between a privileged teacher and a blind student—the privileged teacher can directly pick up the object with vision, while the student policy needs to learn information gathering behavior to pick up the block.

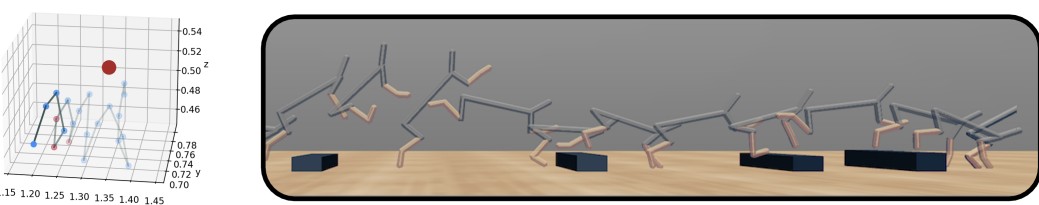

Figure 7: *Scaffolder* discovers spiral search and robust hurdling for blind policies.

The project website showcases several interesting behaviors learned by *Scaffolder* and baselines. *Scaffolder* behaviors broadly fall into two categories depending on the task. Sometimes, it performs information-gathering strategies - in Blind Picking, *Scaffolder* performs spiral search over the workspace to find the block from touch alone. At other times, it acquires robust behaviors invariant to unobservables - in Blind Locomotion, *Scaffolder* discovers robust run-and-jump maneuvers to minimize collisions with unseen randomized hurdles. See Figure 7 for visualizations.

### 4.3 ABLATIONS AND ANALYSIS OF *Scaffolder*

As evidenced above, *Scaffolder* works well, but why? We first replace each privileged component with a non-privileged counterpart to assess component-wise contributions. "No Scaff. WM" opts for training the policy in the target world model over training in the privileged world model. "No Scaff. Critic" uses an unprivileged critic $v_\psi^-(s^-)$ in $R_t^\lambda$ for policy learning. "No Scaff. Explore" collects data with only the target policy. For "No Scaff. Repr.", the representation is trained to reconstruct target observations instead of privileged observations.

Figure 8 compares these ablations on 4 tasks. Overall, all ablations perform poorly relative to *Scaffolder*, indicating the importance of each component. Interestingly, different scaffolded components

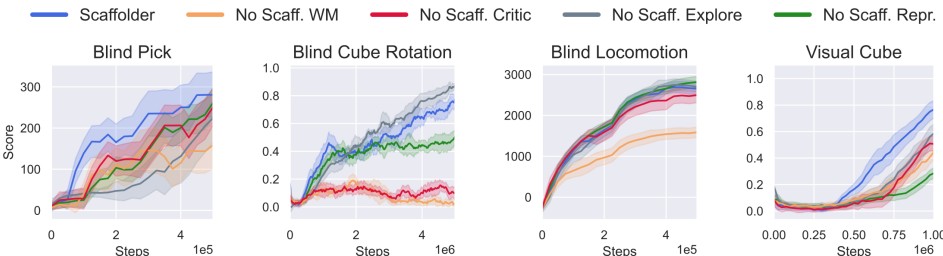

Figure 8: We ablate *Scaffolder* by replacing components with their non-privileged counterparts. All components are important, and the combined method performs best.

prove crucial for different tasks: exploration for Blind Pick, critic for Blind Cube Rotation, world model for Blind Locomotion, and representation for RGB Cube Rotation.

These task-specific trends may be due to the task diversity in S3. In Blind Pick, a blind policy struggles to locate the block, but the vision-based exploration policy can easily locate the block and populate the buffer with useful data. In Blind Cube Rotation, the privileged critic, with access to object pose, can more easily estimate the object-centric reward. Blind Locomotion involves navigating unexpected hurdles, therefore a scaffolded world model, with access to vision, can accurately predict future hurdles. Lastly in Visual Cube, the target policy must encode high-dimensional images, and the scaffolded representation objective encourages encoding of essential object pose and touch information. In all these cases, the combined *Scaffolder* method benefits from cohesively integrating these many routes for privileged sensing to influence policy learning, and performs best.

***Scaffolder* Improves RL Training Signals.** We notice in Figure 8 that dropping the scaffolded WM and its corresponding scaffolded value significantly impact learning. Recall that these components are critical to estimate the policy return (Equation (1)), whose gradients directly determine policy updates. We now examine these policy return estimates.

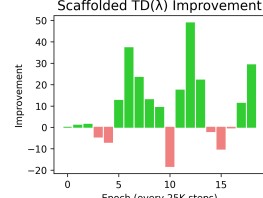

Let the scaffolded TD($\lambda$) return be Equation (1), and the impoverished "target" TD($\lambda$) return be the original DreamerV3 policy objective, which substitutes the blue terms in Equation (1) with target dynamics $p_\phi^-(z_{t+1}^- \mid z_t^-, a_t)$, target rewards $p_\phi^-(r_t^- \mid z_t^-)$, and target critic $v_\psi^-(z_t^-)$.

Figure 9: Comparing TD($\lambda$) estimates.

We compute the mean absolute error (MAE) for each of these two estimates against the ground truth Monte-Carlo return $R_t$. In Blind Pick, every 25K steps while training, we collect $\sim$1000 transitions with the policy and compute $R_t$.

Figure 9 plots Improvement = (target return MAE − scaffolded return MAE). The improvements are mostly positive, indicating that scaffolded TD($\lambda$) is more accurate. This provides evidence for why *Scaffolder* trains better policies than non-privileged counterparts — its policy objective better approximates the true expected return, giving the policy optimizer a better training signal. We find similar trends in other environments in Appendix F.

## 5 CONCLUSION

We have studied how additional observation streams at training time can aid skill learning, introducing a diverse S3 task suite to aid this study. *Scaffolder* successfully exploits privileged sensors across these tasks to improve reinforcement learning signals and train better policies. While these results are promising, there is much room for future work. See Appendix B for extended limitations and future work. Empirically, our task suite focuses on simulated robotic tasks, but other domains like real-world robotics and video games, each with their own forms of privileged sensing, warrant study. For real robots, it may be practical to *estimate* rewards through their sensors (Schenck & Fox, 2017; Fu et al., 2018; Haldar et al., 2023) rather than assume black-box rewards from supervising humans. Here, we expect that improved reward estimates through privileged sensors would offer further advantages (see Appendix G). Our work also opens a window to intriguing deeper questions about the relationship between sensing and learning that it only begins to address: what environment information must a learning agent sense in order to perform a task, and how does this vary with tasks and learning phases? We believe that our findings will prove useful for such future research.

## 6 ACKNOWLEDGEMENTS

This work was supported by the NSF CAREER Award 2239301 and ONR award N00014-22-1-2677. The authors would like to thank the anonymous reviewers for their constructive feedback and members of the GRASP lab and PAL group for their support.

## 7 REPRODUCIBILITY STATEMENT

To ensure reproducibility, we will release all code about *Scaffolder*, baselines, and Sensory Scaffolding Suite on the project website: `https://penn-pal-lab.github.io/scaffolder/`. Our supplementary has a comprehensive overview of our method in Appendix C, experimentation protocol / resource usage / hyperparameters in Appendix E, baseline details in Appendix E.3, and environmental descriptions in Appendix E.4.

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

# A    EXTENDED RELATED WORK

For the sake of brevity, we gave a broad high level overview of related works on sensory scaffolding. We now give a more extensive overview.

First, the sensory scaffolding problem is related to the areas of domain adaptation, domain transfer, and sim-to-real transfer. Such works attempt to transfer an agent trained in a source MDP to a target MDP, differing in observation space, action space, dynamics, rewards, or initial state distribution (i.e. resets) (Taylor et al., 2007; Tobin et al., 2017; Chebotar et al., 2018; Hu et al., 2022; Zhang et al., 2021; Peng et al., 2018; Chen et al., 2023). Sensory scaffolding can be thought of as a special case of observation space transfer, where the source observation space is a superset ($\{o^-, o^p\}$) of the target observation space in a POMDP setting. In contrast, prior work (Sun et al., 2022; Tatiya et al., 2023) focusing on observation transfer typically assume a mapping between the observation spaces to facilitate transfer, i.e. the target image observation space can recover the source simulator state (Pinto et al., 2018), and are formulated with MDPs in mind.

Next, the field of privileged imitation learning is related, but not identical, to sensory scaffolding (i.e. privileged reinforcement learning). In the privileged IL setup, one assumes access to a pre-existing expert policy with privileged access to information, and the objective for the partially observed student policy is to imitate the expert policy (Swamy et al., 2022; Weihs et al., 2021). In sensory scaffolding, we study how privileged sensory streams at training time improve RL policy search for a target policy with diminished senses. We do not assume access to any existing target policy, and the target policy's objective is to maximize reward, not imitate a privileged policy.

# B    EXTENDED LIMITATIONS AND FUTURE WORK

*Scaffolder* trains an additional scaffolded world model and exploration policy on top of the original world model / policy, which adds a sizable amount of compute and resource burden. Training a single dynamics model and policy that is shared across both $o^+$ and $o^-$ would reduce this burden, and we believe using ideas from the multi-task learning literature to be a good first step.

Through our ablations and analysis, we have a good empirical understanding of how *Scaffolder* benefits from privileged observations. On the theoretical side, there is work characterizing privileged imitation learning (Swamy et al., 2022), critics (Baisero & Amato, 2021), world models (Lambrechts et al., 2023) and representation learning (Vapnik & Vashist, 2009), but a concrete theoretical understanding of how privileged information affects the entire RL process and components, is is lacking.

On the practical side, while *Scaffolder* uses privileged observations to improve multiple training-time only RL mechanisms, future work can investigate better ways of leveraging privileged observations. For example, the use of $o^+$ as a reconstruction target for representation learning can likely be replaced with a better privileged representation learning objective.

Next, using additional sensors at training time brings in practical problems. More sensors usually results in increased computation and bandwidth requirements. Furthermore, many multimodal datasets are incomplete, i.e. certain modalities may be missing in subsets of the data.

# C    METHOD DETAILS

Here, we provide a more in-depth exposition of *Scaffolder* and DreamerV3. To start, we set up DreamerV3. In the main paper, we chose to depart from DreamerV3 notation and world model naming convention for two reasons. First, they use $s$ for learned world model state, while in our POMDP settings, $s$ refers to the true environment state. To avoid confusion with true environment state, we chose to use $z$ instead of $s$ for world model state. Next, they split up world model state $s$ into two components, $h, z$ to more accurately write out the exact computation graph. In the main paper, we defined world model state as one component, $z$ to be more conceptually ergonomic.

Now we default back to the Dreamer notation, and will define *Scaffolder* in terms of DreamerV3 notation to make comparison easier.

## C.1 DREAMERV3

In DreamerV3, the recurrent encoder maps observations $x_t$ to stochastic representations $z_t$. Then, the sequence model, a recurrent network with recurrent state $h_t$, predicts the sequence of representations given past $a_{t-1}$. The world model state is the concatenation of $h_t$ and $z_t$. The world model state is then used to reconstruct observations, predict rewards $r_t$, continuation flags $c_t$.

$$
\text{RSSM} \begin{cases} \text{Sequence model:} & h_t = f_\phi(h_{t-1}, z_{t-1}, a_{t-1}) \\ \text{Encoder:} & z_t \sim q_\phi(z_t|h_t, x_t) \\ \text{Dynamics predictor:} & \hat{z}_t \sim p_\phi(\hat{z}_t|h_t) \end{cases} \\
\begin{aligned} \text{Reward predictor:} & \quad \hat{r}_t \sim p_\phi(\hat{r}_t|h_t, z_t) \\ \text{Continue predictor:} & \quad \hat{c}_t \sim p_\phi(\hat{c}_t|h_t, z_t) \\ \text{Decoder:} & \quad \hat{x}_t \sim p_\phi(\hat{x}_t|h_t, z_t) \end{aligned} \tag{3}
$$

Given the sequence batch of inputs $x_{1:T}$, actions $a_{1:T}$, rewards $r_{1:T}$, and continuation flags $c_{1:T}$, the world model is trained to minimize prediction loss, dynamics loss, and the representation loss.

$$
\mathcal{L}(\phi) \doteq \mathbb{E}_{q_\phi}\left[ \sum_{t=1}^{T}(\beta_{\text{pred}}\mathcal{L}_{\text{pred}}(\phi) + \beta_{\text{dyn}}\mathcal{L}_{\text{dyn}}(\phi) + \beta_{\text{rep}}\mathcal{L}_{\text{rep}}(\phi)) \right]. \tag{4}
$$

We refer interested readers to Hafner et al. (2023) for the particular world model loss definitions. We use the default hyperparameters from DreamerV3 for our method and baselines that use Dreamerv3.

Next, DreamerV3 trains Actor-Critic neural networks in imagination. The actor and critic operate over world model states $s_t = h_t, z_t$, and are defined as:

$$
\begin{aligned} \text{Actor:} & \quad a_t \sim \pi_\theta(a_t|s_t) \\ \text{Critic:} & \quad v_\psi(s_t) \approx \mathbb{E}_{p_\phi, \pi_\theta}[R_t] \end{aligned} \tag{5}
$$

The critic is trained to estimate values on imaginary trajectories generated by executing the actor policy with the learned dynamics. The actor policy is trained to maximize the TD($\lambda$) returns of the imagined trajectories, defined below.

$$
R_t^\lambda \doteq r_t + \gamma c_t\left((1-\lambda)v_\psi(s_{t+1}) + \lambda R_{t+1}^\lambda\right) \qquad R_T^\lambda \doteq v_\psi(s_T) \tag{6}
$$

Now, with the DreamerV3 components set up, we are ready to define *Scaffolder* comprehensively, and using DreamerV3 notation.

## C.2 *Scaffolder*

*Scaffolder* trains two world models, one on scaffolded observations $x^+ = [x^-, x^p]$ and one on target observations $x^-$. The scaffolded world model is defined as:

$$
\text{RSSM} \begin{cases} \text{Sequence model:} & h_t^+ = f_\phi(h_{t-1}^+, z_{t-1}^+, a_{t-1}) \\ \text{Encoder:} & z_t^+ \sim q_\phi(z_t^+|h_t^+, x_t^+) \\ \text{Dynamics predictor:} & \hat{z}_t^+ \sim p_\phi(\hat{z}_t^+|h_t^+) \end{cases} \\
\begin{aligned} \text{Reward predictor:} & \quad \hat{r}_t \sim p_\phi(\hat{r}_t|h_t^+, z_t^+) \\ \text{Continue predictor:} & \quad \hat{c}_t \sim p_\phi(\hat{c}_t|h_t^+, z_t^+) \\ \text{Decoder:} & \quad \hat{x}_t^+ \sim p_\phi(\hat{x}_t^+|h_t^+, z_t^+) \end{aligned} \tag{7}
$$

The target world model is defined as:

$$
\text{RSSM} \begin{cases} \text{Sequence model:} & h_t^- = f_\phi(h_{t-1}^-, z_{t-1}^-, a_{t-1}) \\ \text{Encoder:} & z_t^- \sim q_\phi(z_t^-|h_t^-, x_t^-) \\ \text{Dynamics predictor:} & \hat{z}_t^- \sim p_\phi(\hat{z}_t^-|h_t^-) \end{cases} \\
\begin{aligned} \text{Reward predictor:} & \quad \hat{r}_t \sim p_\phi(\hat{r}_t|h_t^-, z_t^-) \\ \text{Continue predictor:} & \quad \hat{c}_t \sim p_\phi(\hat{c}_t|h_t^-, z_t^-) \\ \text{Decoder:} & \quad \hat{x}_t^- \sim p_\phi(\hat{x}_t^-|h_t^-, z_t^-) \end{aligned} \tag{8}
$$

The world models are trained as follows: We sample a trajectory containing observations $x_{1:T}^+$, actions $a_{1:T}$, rewards $r_{1:T}$, and continuation flags $c_{1:T}$. We then train the scaffolded world model as normal and then convert $x_{1:T}^+$ to $x_{1:T}^-$ by simply leaving out privileged observations $x_{1:T}^p$ from the data before feeding it to the target world model.

**Target Actor and Privileged Critic**   Next, we define the target actor and critic. The target actor operates over target world model state $s_t^- = \{h_t^-, z_t^-\}$. Crucially, the critic operates over both the *scaffolded world model state $s_t^+ = h_t^+, z_t^+$* and the target world model state $s_t^-$. Note $s_t^-$ is still required because $s_t^-$ contains info about the historical beliefs of the agent, which is not predictable from only the Markovian learned world model state $s_t^+$. See (Baisero & Amato, 2021) for more information.

The additional environmental information in $s_t^+$ potentially makes value estimation easier for the privileged critic. Furthermore, the critic is trained on trajectories generated by the scaffolded world model, which should better approximate true dynamics. We highlighted in blue the places where privileged components are used instead of their original counterparts.

$$\begin{aligned} \text{Target Actor:} \qquad & a_t \sim \pi_\theta^-(a_t|s_t^-) \\ \text{Privileged Critic:} \qquad & v_\psi(s_t^-, s_t^+) \approx \mathbb{E}_{p_\phi^+, \pi_\theta^-}[R_t] \end{aligned} \qquad (9)$$

Now, the actor is trained with to maximize TD($\lambda$) returns $R_t^\lambda$ of generated trajectories. We improve all components within this objective with scaffolded observations, and highlight them in blue.

First, the policy objective is written as:

$$\mathcal{L}(\theta) \doteq \sum_{t=1}^T \mathbb{E}_{\pi_\theta^-, p_\phi^+}[R_t^\lambda / \max(1, S)] - \eta \mathrm{H}[\pi_\theta^-(a_t|s_t^-)] \qquad (10)$$

where we start by using the scaffolded world model to generate imaginary trajectories, using the Nested Latent Imagination procedure described in Section 3.

We further incorporate scaffolded components into the TD($\lambda$) return. Because we have access to $s_t^+$ from generating trajectories in the scaffolded world model, we can use the scaffolded reward, continue, and critic to compute TD($\lambda$) return.

$$R_t^\lambda \doteq r_t + \gamma c_t\Big((1-\lambda)v_\psi(s_{t+1}^-, s_{t+1}^+) + \lambda R_{t+1}^\lambda\Big) \qquad R_T^\lambda \doteq v_\psi(s_T^-, s_T^+) \qquad (11)$$

**Additional Exploration Policy**   We train an additional exploration policy $\pi^e(a_t \mid s_t^+)$ that operates over scaffolded world model state during training time, to collect informative trajectories. The intuition is that with better sensing, $\pi^e$ can solve the task more quickly and gather relevant data. To do so, we simply define separate actor critic networks for the exploration policy, that depend on $s^+$. It is trained to maximize reward using standard imagination within the scaffolded world model.

$$\begin{aligned} \text{Exploration Actor:} \qquad & a_t \sim \pi_\theta^e(a_t|s_t^+) \\ \text{Exploration Critic:} \qquad & v_\psi^e(s_t^+) \approx \mathbb{E}_{p_\phi^+, \pi_\theta^e}[R_t] \end{aligned} \qquad (12)$$

We alternate between collecting episodes with the exploration policy and target policy in a 1:1 ratio.

**Target Policy Representation**   We modify the target decoder to $p_\phi(\hat{x}_t^+ \mid h_t^-, z_t^-)$ so that it enforces the representation to be more informative of the privileged observations, which is potentially useful for control.

## C.3   NESTED LATENT IMAGINATION

Here, we motivate and explain the nested latent imagination procedure more in detail. First, let's start with a brief overview of DreamerV3's latent imagination procedure.

Recall that DreamerV3 uses the world model components (dynamics, posterior) for two roles: 1) as a environment simulator to generate synthetic experience for policy improvement, and 2) as a latent state encoder that aggregates observation history for the policy.

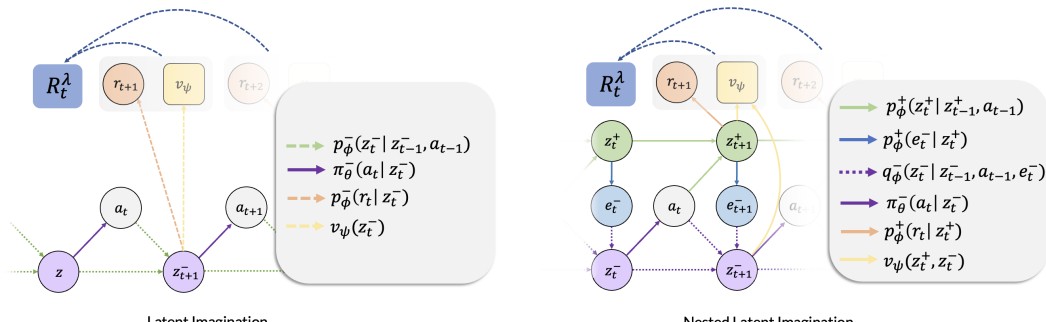

Figure 10: Left: DreamerV3's Latent Imagination - note that it only uses impoverished, target dynamics, rewards, and values to generate trajectories for policy improvement. Right: *Scaffolder* uses scaffolded dynamics, rewards and values.

Concretely, the dynamics $p^-(z_{t+1}^- \mid z_t^-, a_t)$ is the environment simulator, and the posterior $p^-(z_t^- \mid z_{t-1}^-, a_{t-1}, e_t^-)$ encodes present $e_t^-$ and history $z_{t-1}^-, a_{t-1}$ into a compact latent state $z_t^-$.

Referring to the left side of Figure 10, DreamerV3's latent imagination generates trajectories by repeating the following three steps.

1. **Policy Inference:** At timestep $t$, the latent state $z_t^-$ is fed into the target policy $\pi^-(\cdot \mid z_t^-)$ to generate an action $a_t$.

2. **Dynamics Inference:** The current latent state and action $(z_t^-, a_t)$ are fed into the dynamics $p^-(z_{t+1}^- \mid z_t^-, a_t)$ to generate a future state $z_{t+1}^-$.

3. **Credit Assignment:** With the future state as input, rewards $p^-(r_{t+1} \mid z_{t+1}^-)$ and value estimates $v(z_{t+1}^-)$ are computed.

After running these 3 steps for $H$ timesteps, a trajectory $\tau = \left[z_1^-, a_1, z_2^-, \ldots z_H^-\right]$ is created. The policy is trained to maximize the TD($\lambda$) return of all the trajectories, which is a function of the dynamics, rewards, and values estimates.

Consider what happens to this procedure if the target observations are extremely impoverished, i.e. have a high loss of information from the scaffolded observations. Then the target dynamics model $p^-(z_{t+1}^- \mid z_t^-, a_t^-)$, rewards $p^-(r_{t+1} \mid z_{t+1}^-)$ and value estimates $v(z_{t+1}^-)$ will be inaccurate.

Instead, we propose to train an additional scaffolded world model, that can take advantage of the scaffolded observations to learn a more accurate environmental simulator for generating trajectories, replacing the role of the target dynamics in DreamerV3. However, because we still need to run the target policy, we retain and train the target world model components and use the target posterior to encode latent states from target observations.

Following the right side of Figure 10:

1. **Policy Inference:** At timestep $t$, we have the target latent state $z_t^-$ and the scaffolded latent state $z_t^+$. The target latent is fed into the target policy $\pi^-(\cdot \mid z_t^-)$ to generate an action $a_t$.

2. **Scaffolded Dynamics Inference:** The current scaffolded latent state and action $(z_t^+, a_t)$ are fed into the scaffolded dynamics $p^+(z_{t+1}^+ \mid z_t^+, a_t)$ to generate a future state $z_{t+1}^+$.

3. **Scaffolded Credit Assignment:** With the scaffolded future state as input, rewards $p^+(r_{t+1} \mid z_{t+1}^+)$ and value estimates $v(z_{t+1}^+, z_{t+1}^-)$ are computed.

4. **Target State Inference:** At this point, we have the future scaffolded state $z_{t+1}^+$, but the policy requires $z_{t+1}^-$ to start step 1 for the next iteration.

   We can't directly convert or learn a mapping $z^+ \rightarrow z^-$, since they encode fundamentally different information (see Baisero & Amato (2021)). Instead, we can learn a mapping $z^+ \rightarrow o^-$, since $z^+ \rightarrow o^+$ via DreamerV3's reconstruction objective, and $o^+$ is a superset of $o^-$.

By training the target embedding predictor $p^+(e^- \mid z^+)$ where $e^-$ is the low-level embedding of the target observation, we can now infer the target latent state $z_t^-$ with the target posterior $q(z_{t+1}^- \mid z_t^-, a_t, e_t^-)$. With $z_{t+1}^-$ from the target posterior, we now can start step 1 for the next iteration.

After running these 4 steps for $H$ timesteps, a trajectory $\tau = \left[ z_1^-, z_1^+, a_1, z_2^-, z_2^+ \ldots z_H^-, z_H^+ \right]$ is created. The policy is trained to maximize the TD($\lambda$) return of all the trajectories, which is a function of the scaffolded dynamics, rewards, and values estimates. Later, we show that these scaffolded TD($\lambda$) estimates using scaffolded components are more accurate than the non-privileged TD($\lambda$) estimates, explaining why *Scaffolder* trains better policies than DreamerV3.

# D    RUNTIME AND RESOURCE COMPARISON

Our method builds off DreamerV3, so it is compute efficient, requiring only 1 GPU and 4 CPUs for each run. Similarly, baselines building off of DreamerV3 like Informed Dreamer, DreamerV3+BC, Guided Observability also are compute efficient. We train on Nvidia 2080ti, 3090, A10, A40, A6000, and L40 GPUs. In contrast, model-free methods like RMA and AAC require large batches for PPO / Actor-Critic training, requiring 128 CPU workers to have fast and stable training.

Next, we give the runtime in hours for each method, for all experiments in Figure 11. Note that these are approximate, since we run methods over different types of GPUs and CPUs. In general, *Scaffolder* takes $\sim 30\%$-$40\%$ longer to run than DreamerV3 due to training an additional world model and exploration policy, but *Scaffolder* is much more sample efficient and reaches higher performance with less environment steps, making up for slower wall time. Similarly, DreamerV3+BC is also slower compared to DreamerV3 because it trains the teacher policy and teacher world model alongside the student policy and world model. In general, RMA and AAC take at least 1 day to train to reach 100M steps.

We plot the learning curves over hours rather than environment steps in Figure 12. The trends are largely the same as the learning curves over environment steps in Figure 6. This is because *Scaffolder* is much more sample-efficient than baselines, reaching higher performance earlier in walltime even if it takes more time per-step to train the additional models.

| Env. | *Scaffolder* | Informed Dreamer | DreamerV3+BC | Guided Obs. | RMA+(100M) | AAC(100M) | DreamerV3 |
|---|---|---|---|---|---|---|---|
| Blind Pick | 18 | 14 | 18 | 14 | 36 | 36 | 14 |
| Blind Locomotion | 16 | 12 | 24 | 16 | 24 | 24 | 16 |
| Blind Deaf Piano | 18 | 14 | 18 | 14 | 36 | 36 | 14 |
| Blind Pen | 12 | 9 | 12 | 9 | 36 | 36 | 9 |
| Blind Cube | 12 | 9 | 12 | 9 | 36 | 36 | 9 |
| Noisy Monkey | 20 | 12 | 20 | 12 | 36 | 36 | 12 |
| Wrist Pick-place | 22 | 16 | 22 | 16 | 36 | 36 | 16 |
| Occluded Pick-Place | 9 | 5 | 9 | 5 | 36 | 36 | 5 |
| RGB Pen | 3 | 2 | 3 | 2 | 36 | 36 | 2 |
| RGB Cube | 3 | 2 | 3 | 2 | 36 | 36 | 2 |

Figure 11: Approximate runtime in hours for each method, over the tasks.

# E    EXPERIMENT DETAILS

## E.1    EVALUATION PROTOCOL

In the main results in Figure 6, we report the score for each task. The score is a task-specific metric of performance. We use return (sum of rewards) by default. The Pen / Cube rotation tasks use success rate and Noisy Monkey uses number of handholds. While the policy trains, we periodically evaluate the policy every 15000 training steps, and log the mean score over 15 evaluation episodes. We launch 4-10 seeds for each method, and report median and standard error of the evaluation scores in the learning curves over the seeds. We give each task a sample budget ranging from 250K to 5M.

To compute the final normalized median scores, we perform the following steps for each task. First, obtain a lower and upper bound on performance by running Dreamerv3 using either $o^-$ or $o^+$ as inputs for the task's training budget. Next, we take each method's median score over seeds at the end of training and normalize it by the lower and upper bound scores. We do this for each method.

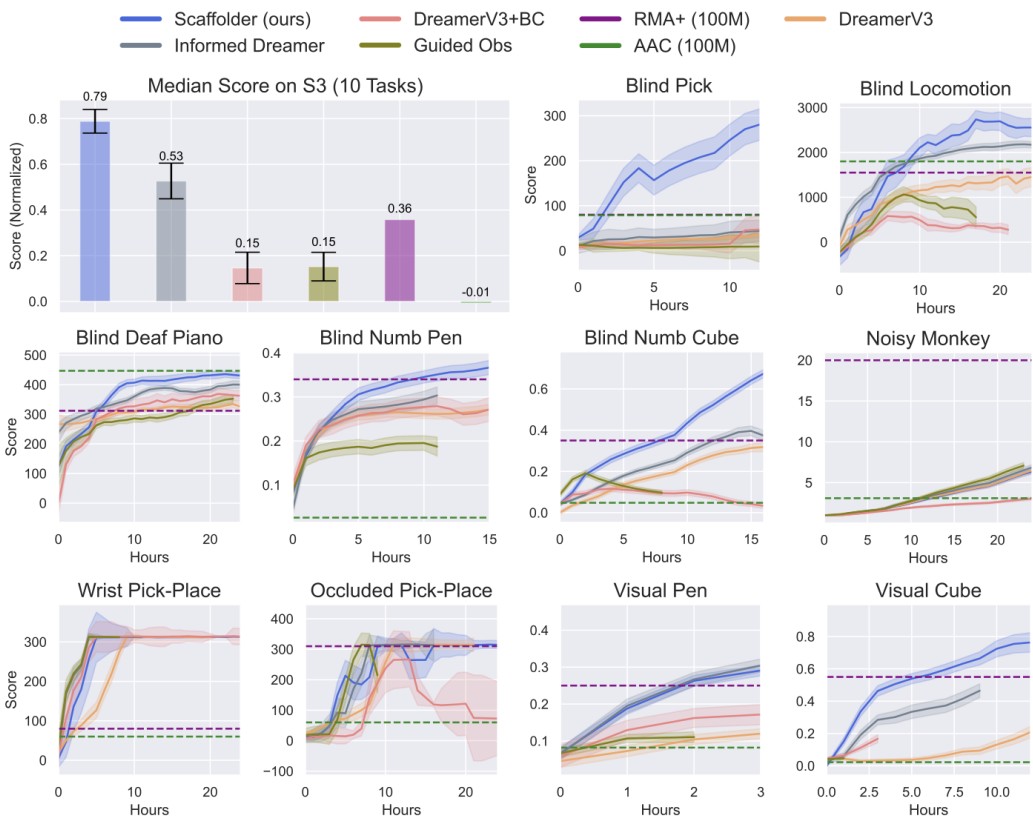

Figure 12: We plot performance over walltime for *Scaffolder* and baselines.

We then average the normalized scores for each method for all tasks, and report this as the final median score.

### E.2 *Scaffolder* HYPERPARAMETERS

Next, we describe the hyperparameter settings for *Scaffolder*. In short, we found hyperparameter search to be short and easy, due to the robustness of the DreamerV3 algorithm. For new environments, we found that DreamerV3 only needs tuning for two hyperparameters, the model size and update to data (UTD) ratio. We follow an easy guideline for tuning these - more complicated dynamics generally require larger models, and tasks with harder exploration require more data and fewer updates (low UTD). As a result, we found hyperparameter settings that work for tasks with similar properties, as seen in Table 1.

We did not tune these two DreamerV3 hyperparameters towards *Scaffolder*- rather, when we used model sizes and UTD ratios from when we ran DreamerV3 with privileged inputs as a reference method used for computing the upper bound scores. These same settings are then used for all DreamerV3 methods (*Scaffolder*, Informed Dreamer, DreamerV3+BC, Guided Observability) to be consistent.

Table 1: DreamerV3 hyperparameters

| Model, UTD | |
| --- | --- |
| Small, 512 (Simple Dynamics, Easy Exploration) | Blind Pick, Blind Locomotion, Wrist Pick-Place, Occluded Pick-Place |
| Large, 512 (Complex dynamics, Easy Exploration) | Blind Deaf Piano |
| Large, 16 (Complex Dynamics, Hard Exploration) | Noisy Monkey, Blind Pen, Blind Cube, RGB Pen, RGB Cube |

### E.3 BASELINES

**DreamerV3**  We run DreamerV3 "out of the box" with only access to target observations for all tasks. See the prior hyperparameters discussion for details.

**Informed Dreamer**  Informed Dreamer adds an additional loss term to DreamerV3 that pushes the world model to decode privileged information from the target observations. Informed Dreamer works best when the privileged information is predictable from the impoverished, target observation. Due to the simplicity of the method, which just decodes additional observations, we can just use the DreamerV3 codebase directly to run Informed Dreamer by changing the decoder outputs.

**DreamerV3+BC**  DreamerV3+BC trains two policies simultaneously, a teacher policy on privileged observations and a student policy on target observations. The teacher policy trains with the same objective as DreamerV3, while the student policy has a combined reinforcement learning and behavior cloning objective that pushes the student to both maximize return and mimic the teacher's actions. The BC objective assumes the optimal student and teacher have the same action distributions for it to be helpful.

This assumption does not always hold true and is violated in multiple of S3's environments, including Wrist Camera Pick-and-Place, Occluded Pick-and-Place, and Blind Pick. The student's RL objective can help with this imitation gap, but DreamerV3+BC is still outperformed by numerous other methods that exploit privileged information at training time. We implement DreamerV3+BC on top of DreamerV3.

To find the best balance between RL and BC objective for each task, we try BC weighting values of $0.1, 0.01, 0.001$ for all 10 tasks, and report DreamerV3+BC results with the best weight for each task.

**Guided Observability**  Guided Observability implements a schedule for removing privileged information during training. The policy is initially given both privileged and target observations, but as training progresses, the privileged information is dropped out with increasing probability. This dropout probability linearly increases with training iterations until only the target observations remain. We used a 50% cutoff for access to privileged information for all tasks, but this hyperparameter can realistically be tuned for each environment.

Guided Observability is algorithm agnostic but was implemented on top of DreamerV3 for this baselines so as to remain as comparable as possible with Scaffolder and other Dreamer-based baselines. Our implementation closely mimicked PO-GRL (Weigand et al., 2021), but had one small modification. In PO-GRL, dropped-out observations are directly placed in the replay buffer. For our implementation, however, only scaffolded observations ($o^+$) are placed in the replay buffer, whether they were dropped out during the data collection process or not. Privileged information sampled from the replay buffer for world model training is then probabilistically dropped out on the same training schedule. This mitigates the risk of privileged information leaking into updates late in the training process. We choose a simple linear annealing schedule that anneals towards 0 by 50% of training for all tasks.

**RMA**  RMA shares the same policy between teacher and student, and assumes that the privileged teacher training uses low-dimensional privileged system information as input. Therefore, it is not conceptually or practically easy to incorporate noisy, high dimensional observations into teacher training. So instead, we just give RMA an advantage by giving it access to privileged low dimensional states instead of privileged observations. We use the RMA codebase from (Qi et al., 2023) to run our experiments.

**Asymmetric Actor Critic (AAC)**  Asymmetric Actor Critic (Pinto et al., 2018) conditions the critic on scaffolded observations, while only conditioning the actor on target observations. AAC assumes that providing privileged information to the critic will allow it to better estimate the value of states and thus, improve policy learning.

AAC was implemented on top of cleanrl's (Huang et al., 2022b) implementation of PPO. To account for the relative sample inefficiency of PPO and model-free algorithms in general, a 100M sample budget was given to AAC for each task.

### E.4 SENSORY SCAFFOLDING SUITE

**Blind Pick**  A two-fingered robot arm must pick up a randomly initialized block using only proprioception (gripper position) and touch sensing. During training, it can use multiple privileged RGB camera inputs (two static cameras and one wrist camera).

- Observation Space:
    - $o^p$: Privileged observations include three low-resolution RGB cameras: two static cameras face the workspace from the front and side, while one egocentric camera is located in the wrist.
    - $o^-$: The target observation space includes the gripper's position, velocity, and open/close state. Additionally, the robot has two binary touch sensors on each gripper finger.
- Action Space: Gripper velocity and gripping force.
- Reward: Task reward for successfully picking up the block and auxiliary rewards computed using distances between the gripper, block, and picking goal positions.

**Blind Locomotion**  In the Half Cheetah task, a proprioceptive policy must run and jump over randomly sized and positioned hurdles. During training, privileged RGB images of the Half Cheetah and its nearby hurdles are given, allowing the agent to see obstacles before bumping into them (see Figure 2).

- Observation Space:
    - $o^p$: RGB camera that tracks the Half Cheetah as it moves.
    - $o^-$: Joint angles and velocities.
- Action Space: Torques for each of the 6 moveable joints on the Half Cheetah.
- Reward Function: Reward proportional to the Half Cheetah's forward-moving velocity with a penalty for incurred control costs.

**Blind and Deaf Piano**  A pair of 30-DoF Shadowhands in the Robopianist simulator (Zakka et al., 2023) must learn to play "Twinkle Twinkle Little Star" using only proprioception. At training time, the policy has access to future notes, piano key presses, and suggested fingerings, which emulates having vision to see sheet music and hearing to determine which keys were pressed.

- Observation Space:
    - $o^p$: Future notes for the next ten control steps, suggested fingerings for each set of notes, current piano key presses, previous action, and previous reward.
    - $o^-$: Joint angles and forearm position for each Shadowhand.
- Action Space: Desired joint angles and forearm position for each Shadowhand.
- Reward Function: Task reward for playing correct notes with a penalty incurred for incorrect notes. Additionally, auxiliary rewards are provided to encourage the fingers to stay close to the keys and minimize energy.

**Blind and Numb Pen**  A proprioceptive policy must control a 30-DoF Shadowhand to rotate a pen from a randomized initial orientation to a randomized desired orientation. The target policy receives joint angles, the initial orientation of the pen, and the desired goal orientation. During training, the policy has access to the object pose and touch sensors.

- Observation Space:
    - $o^p$: Pen pose and dense touch sensing.
    - $o^-$: Joint angles, initial pen pose, goal pen pose.

- Action Space: Joint angles.
- Reward Function: Dense positive reward proportional to the similarity between the current and target pen orientations. A negative penalty is incurred for dropping the pen.

**Blind and Numb Cube**   Similar to Blind and Numb Pen rotation, a proprioceptive policy must control a 30-DoF Shadowhand to rotate a cube from a randomized initial orientation to a randomized desired orientation.

- Observation Space:
    - $o^p$: Cube pose and touch sensing.
    - $o^-$: Joint angles, initial cube pose, goal cube pose
- Action Space: joint angles.
- Reward Function: Dense positive reward proportional to the similarity between the current and target cube orientations. A negative penalty is incurred for dropping the cube.

**Noisy Monkey Bars**   A 13-link gibbon must swing between a fixed set of handholds in a 2D environment using the brachiation simulator (Reda et al., 2022). Handholds are placed far enough apart such that the gibbon must use its momentum to swing and fly through the air to reach the next handhold. To simulate imperfect sensors on a robotic platform, Gaussian noise is added to the target observations, while privileged observations represent true simulator states. To improve policy learning in this difficult control task, the relative position of the model's center of mass from a provided reference trajectory is used as an auxiliary reward.

- Observation Space:
    - $o^p$: Privileged observations include true simulator states, including the gibbon model's overall velocity and pitch, $\sin$ and $\cos$ of each joint angle, joint velocities, height of the current arm from the model's center of mass, and whether each hand is currently grabbing or released. Additionally, the relative position to the next handhold and the relative distance to the reference trajectory are provided to the policy.
    - $o^-$: The target observation space is equivalent to the privileged observation space but with added Gaussian noise. Gaussian noise is applied to the true environment states rather than the observations to more accurately represent noisy sensors. For example, noise is added to joint angles rather than the $\sin$ and $\cos$ of those angles that the policy receives.
- Action Space: Desired joint angle offsets for each joint on the gibbon model and a binary grab/release action for each hand.
- Noise: Gaussian noise with $0$ mean and $0.05$ standard deviation is independently applied to each simulator state component. Angles are measured in radians and distance is measured in meters. For reference, the gibbon is $0.63$m tall.
- Reward Function: The reward function includes three terms: a style term to encourage the model to stay relatively upright and minimize jerky motions, a tracking term to encourage the model's center of mass to stay close to the provided reference trajectory, and a sparse task reward term for each additional handhold reached.

**Wrist Camera Pick-and-Place**   This task examines the impact of multiple privileged 3rd-person cameras on learning a target active-perception policy with wrist camera input. Given the wrist camera's restricted field of view, the target policy must concurrently learn to move to locate the block and bin while executing the task. This task mirrors real-world scenarios such as mobile manipulation, where a controllable wrist camera with a limited field of view might be the sole visual sensor.

- Observation Space:
    - $o^p$: Two static, low-resolution RGB cameras facing the workspace from the front and side.
    - $o^-$: Proprioceptive information including the gripper's position, velocity, and open/close state, touch sensing from two binary touch sensors located on each gripper finger, and vision from a low-resolution RGB egocentric camera located in the wrist of the robot.
- Action Space: Gripper velocity and force.

- Reward Function: Sparse task reward for placing the block in the bin and auxiliary rewards computed using distances between the gripper, block, and bin positions.

**Occluded Pick-and-Place**  Rather than employing active visual perception at test time, this task examines the impact of active-perception as privileged sensing at training time. The target policy must use an RGB camera from an occluded viewpoint alongside proprioception and touch sensing to pick up a block behind a shelf and place it into a bin. Both object and bin locations are randomly initialized. During training time, the robot gets access to a privileged wrist mounted camera, enabling it to perform active visual perception to locate the block during training (see Figure 2).

- Observation Space:
    - $o^p$: Two low-resolution RGB cameras: one static, unoccluded camera facing the workspace from the side and one active-perception camera located in the wrist of the robot.
    - $o^-$: One occluded low-resolution RGB camera. Although this camera does not reveal any information about the block, it does encode the position of the randomly placed bin.
- Action Space: Gripper velocity and force.
- Reward Function: Sparse task reward for placing the block in the bin and auxiliary rewards computed using distances between the gripper, block, and bin positions.

**Visual Pen Rotation**  We carry over the setup from the previous two blind and numb dexterous manipulation tasks with privileged object pose and contact sensors. In this task, however, the target policy gains access to a top-down RGB image (see Figure 5) to rotate a pen from a randomized initial orientation to a randomized desired orientation. At training time, the policy has access to the pen pose and touch sensing.

- Observation Space:
    - $o^p$: Pen pose and dense touch sensing.
    - $o^-$: Joint angles, One top-down, low-resolution RGB camera. desired pen pose
- Action Space: Desired joint angles.
- Reward Function: Dense positive reward proportional to the similarity between the current and target pen orientations. A negative penalty is incurred for dropping the pen.

**Visual Cube Rotation**  Similarly to Visual Pen Rotation, a visual target policy conditioned on a top-down RGB image and proprioception must rotate a block to a desired orientation.

- Observation Space:
    - $o^p$: Cube pose and dense touch sensing.
    - $o^-$: Joint angles, topdown RGB camera, desired cube pose.
- Action Space: Desired joint angles.
- Reward Function: Dense positive reward proportional to the similarity between the current and target cube orientations. A negative penalty is incurred for dropping the cube.

## F  ADDITIONAL TD($\lambda$) ERROR EXPERIMENTS.

Let the scaffolded TD($\lambda$) return be Equation (1), and the impoverished "target" TD($\lambda$) return be the original DreamerV3 policy objective, which substitutes the blue terms in Equation (1) with target dynamics $p_\phi^-(z_{t+1}^- \mid z_t^-, a_t)$, target rewards $p_\phi^-(r_t^- \mid z_t^-)$, and target critic $v_\psi^-(z_t^-)$. We compute the mean absolute error (MAE) for each of these two estimates against the ground truth Monte-Carlo return $R_t$.

We evaluate this in two additional environments, Blind Navigation and Blind Locomotion. In Blind Navigation, a randomly initialized agent must find a randomly initialized goal point in a 15x15 gridworld. The privileged observation is the location of the goal in every epsiode, while the target observation is the x,y location of the agent. Every 25K steps while training, we collect ~1000 transitions with the policy and compute $R_t$.

Figure 9 plots Advantage = (target return MAE − scaffolded return MAE). If the target error is higher than the scaffolded estimate's error, then Advantage is positive. The advantages are mostly positive, indicating that scaffolded TD($\lambda$) is more accurate. This provides evidence for why *Scaffolder* trains better policies than non-privileged counterparts — its policy objective better approximates the true expected return, giving the policy optimizer a better training signal.

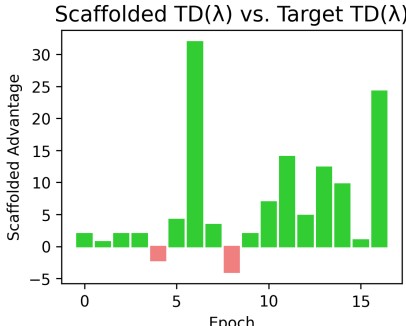

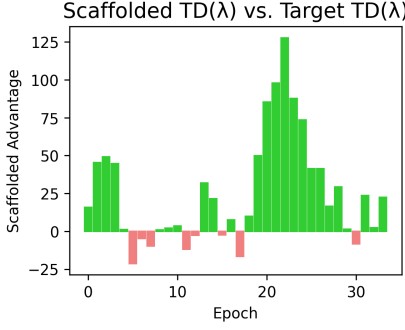

Figure 13: Top: Blind Nav., Bottom: Blind Locomotion

## G  SCAFFOLDER WITH REWARDS ESTIMATED FROM PRIVILEGED SENSORS

To simulate realistic real world RL scenarios, we estimate reward signals from noisy sensors rather than using rewards computed from simulator state. We modify the Blind Pick environment's reward function to use estimated object pose rather than ground truth object pose, acquired via color-segmentation from the privileged cameras. The estimated object pose now has noise due to partial occlusions of the block by the robot gripper.

We train Scaffolder and other methods on the estimated reward, and evaluate on the ground truth return. As seen in Figure 14, all methods suffer some performance degradation. *Scaffolder* gets around 200 return whereas in the original Blind Pick it gets 300. However, the performance trends are consistent. Looking at policy rollouts, we find that only *Scaffolder* can reliably pick up the block and all other methods fail.

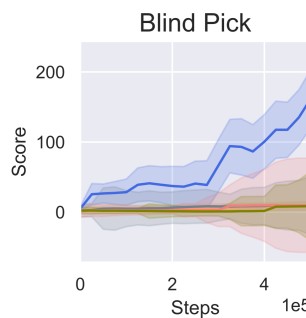

Figure 14: *Scaffolder* still outperforms baselines in the estimated reward setting.

## H  *Scaffolder* ON PRE-EXISTING ENVIRONMENTS

We ran *Scaffolder* and DreamerV3 (operating only on target input, consistent with our other results) on two other environments from the COSIL Nguyen et al. (2022) paper, Bumps-2D and Car-Flag.

In Bumps-2D, a robot arm must find the bigger bump out of a pair of randomly initialized bumps in a 2D grid, using only proprioception and touch sensing. In Car-Flag, a car using just proprioception must find a randomly initialized green flag. If the car finds a blue flag, the green flag position is revealed. In both environments, the state expert can directly go to the bigger bump / green flag, while the student policy must perform information gathering actions to solve the task.

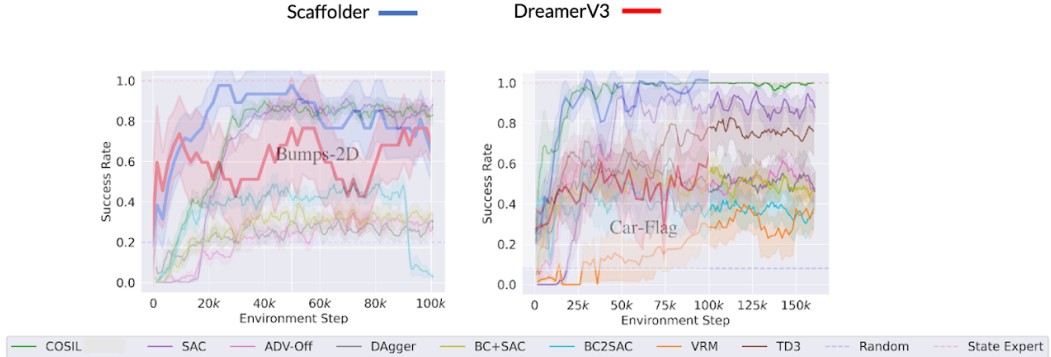

Figure 15: Performance of *Scaffolder* and DreamerV3 on two environments from COSIL.

As seen in Figure 15, the new results are consistent with those reported in our paper on the 10 S3 tasks: *Scaffolder* clearly outperforms DreamerV3, further establishing the versatility of our approach for utilizing privileged sensory information.

In addition, these experiments also enable new comparisons with the prior work evaluated in COSIL. Note that COSIL and its baselines operate in a slightly different setting from *Scaffolder*: they assume access to a perfect, hand-coded scaffolded policy , whereas *Scaffolder* trains both target and scaffolded policies from scratch, and in tandem, starting at environment step 0. This means that, when comparing learning curves, we must remember that COSIL effectively "starts ahead" of *Scaffolder*. Even with this handicap, *Scaffolder* easily outperforms COSIL on Bumps-2D, and matches it on Car-Flag. These results on COSIL's own evaluation environments show further evidence of *Scaffolder* 's improvements over prior state of the art.

