# OpenReview forum: "Privileged Sensing Scaffolds Reinforcement Learning"
_ICLR.cc/2024/Conference — ICLR 2024 spotlight_

### Official Review · Reviewer_Sbzi · 2023-10-27

**Soundness:** 3 good
**Presentation:** 4 excellent
**Contribution:** 4 excellent
**Rating:** 10
**Confidence:** 4

**Summary:**

The authors propose Scaffolder, a model-based RL method that can leverage privileged information at training time. Here, priviledged information means an MDP where:

* There is some true state $s$ with observations $o^+$, where $o^+$ includes privileged info we do not want to assume is available at inference time. (i.e. ground truth object state). This could be used to train a privileged policy $\pi^+$ that cannot be used as-is for inference.
* There is an observation $o^-$ for unprivileged / impoverished target observations, which our final policy $\pi^-$ will depend on.
* We would like the best $\pi^-$ possible while leveraging information in $o^+$.

This problem has been studied in a number of recent works, often in a model free manner. This paper aims to leverage privileged information in a model-based manner.

To do so, the authors train 2 worlds models. The world model subroutine used is DreamerV3. One models privileged information $o^+$ and the other models target information $o^-$. In this summary I'll call them WM+ and WM-.

A "latent translator" is learned to translate WM+ into an observation that WM- can use in its rollouts. Specifically: WM+ has internal latent state $z^+$. We fit a prediction model $p(e^-|z^+)$, where $e^- \approx emb(o^-)$. Part of DreamerV3 is learning a posterior $q(z_{t+1}|z_t,a_t,e_{t+1}=emb(o_{t+1}))$ that infers latent state from history and current observation. By replacing the impoverised $e^- = emb(o^-)$ with a prediction driven by privileged latent $z^+$, we can channel some privileged information into the rollout of $z^-$, assuming that privileged information is eventually observable in unprivileged information.

This latent translator lets us use $\pi^-(a|z^-)$ to rollout both WM+ and WM-, giving a sequence of latents $(z^+,z^-)$ from both world models. The learned reward function is then defined as $R(z^+,z^-)$ to allow observing privileged information in the critic.

We additionally fit a $\pi^+$ directly in the privileged world model, using this solely to generate additional exploratory data (it is possible that some exploration behaviors are easier to learn or discover from privileged information). Last, a decoder is trained to map $z^-$ to $o^+$. To me this seems the least motivated, in that not all parts of $o^+$ should be predictable from $z^-$ in the first place, but it seems to empirically be effective.

The evaluation of Scaffolder is done in a variety of "sensory blindfold" tasks, mostly robotics based, where some sensors are defined as privileged and some are not. The method is compared to DreamerV3 on just target information, a few variants of DreamerV3 based on only fitting one world model with decoding of privileged information, and some model free baselines like slowly decaying use of privileged information, asymmetric actor critic, or using BC to fit an unprivileged policy to a privileged one.

**Strengths:**

The paper provides a good overview of previous methods for handling privileged information, proposes an evaluation suite for studying the problem of privileged information, and proposes a modification of DreamerV3 that handles the information better than Informed Dreamer. There is a significant amount of machinery around Scaffolder, but it's mostly clear why the added components ought to be helpful for better world modeling and policy exploration. The model-free baselines used are pretty reasonable, and it is shown that Scaffolder still outperforms these model free methods even when the model free methods are given significantly more steps.

Finally, the evaluation suite covers a wide range of interesting robot behaviors and the qualitative exploratory methods discovered to handle the limited state (i.e. spiraling exploration behavior) are quite interesting. The S3 suite looks like a promising testbed for future privileged MDP work, separate from the algorithmic results of the paper.

**Weaknesses:**

In some sense, Scaffolder requires doing 2x the world model fitting, as both the z^- and z^+ models need to be fit for the approach to work. In general, this is "fair" for model-based RL, which is usually judged in number of environment interactions rather than number of gradient steps, but it very definitely is a more complex system and this can introduce instability.

A common actor-critic criticism is that the rate of learning between the actor and critic needs to be carefully controlled such that neither overfits too much to the other. Scaffolders seems to take this and add another dimension for the rate of learning between the privileged world model and unprivileged one, as well as the learning speed of the privileged exploratory actor $\pi^+$ and target actor $\pi^-$.

**Questions:**

In general, the paper focuses on the online, tabula rasa case where we are in an entirely unfamiliar environment. How adaptable is this method to either the offline case, or the finetuning case where we have an existing policy / world model?

---

> ### Author Response · Authors · 2023-11-16
> **Author Response**
>
> Thank you for your review!
>
> > Scaffolder requires doing 2x the world model fitting, as both the z^- and z^+ models need to be fit for the approach to work. In general, this is "fair" for model-based RL, which is usually judged in number of environment interactions rather than number of gradient steps, but it very definitely is a more complex system and this can introduce instability. A common actor-critic criticism is that the rate of learning between the actor and critic needs to be carefully controlled such that neither overfits too much to the other. Scaffolders seems to take this and add another dimension for the rate of learning between the privileged world model and unprivileged one, as well as the learning speed of the privileged exploratory actor
>
> We found the hyperparameter tuning process for Scaffolder, and DreamerV3 in general, to be short and easy. On a completely new environment and task, we found that DreamerV3 only needs tuning for two hyperparameters, the model size and the update to data (UTD) ratio. We follow an easy guideline for tuning these - more complicated dynamics generally require larger models, and tasks with harder exploration require more data and fewer updates (low UTD).
>
> We did not tune these two DreamerV3 hyperparameters towards Scaffolder - rather, when we used model sizes and UTD ratios from when we ran DreamerV3 with privileged inputs as a reference method used for computing the upper bound scores. These same settings are then used for all DreamerV3 methods (Scaffolder, Informed Dreamer, DreamerV3+BC, Guided Observability) to be consistent.
>
> As a result, we found hyperparameter settings that work for tasks with similar properties:
>
> - Model / UTD
> - Small / 512: Blind Pick, Blind Locomotion, Wrist Pick-Place, Occluded Pick-Place (Simple Dynamics, Easy Exploration)
> - Large / 512: Blind Deaf Piano (Complex dynamics, Easy Exploration)
> - Large / 16: Noisy Monkey, Blind Pen, Blind Cube, RGB Pen, RGB Cube (Complex Dynamics, Hard Exploration)
>
> No tuning was needed for the actor-critic learning rate, the world model learning rates, and scaffolded / target policy learning rates.
>
> > In general, the paper focuses on the online, tabula rasa case where we are in an entirely unfamiliar environment. How adaptable is this method to either the offline case, or the finetuning case where we have an existing policy / world model?
>
> Thank you for the interesting question. Indeed, we have thought about this too, and believe that Scaffolder may be well-suited for learning from offline data. Many large offline trajectory datasets, such as RT-X [1], have additional privileged information, in the form of metadata, captions, additional modalities, etc. that can be exploited by Scaffolder as privileged sensors to better train the target policy. Training Dreamer style models entirely offline may be challenging, but it may be possible to extend current offline model-based RL frameworks [2,3]  to achieve this.
>
> Scaffolder may also be an excellent fit for finetuning policies in settings where we have some prior knowledge. Scaffolder has explicit components to model dynamics models, policies, value functions, exploration policies etc. This  permits injecting varied types of prior knowledge: for example, the privileged dynamics model could be initialized based on prior knowledge and finetuned. Residual learning methods [4], shown to have success in RL settings before, may also be combined with Scaffolder for further improved finetuning performance."
>
>
> [1] Padalkar, Abhishek, et al. "Open x-embodiment: Robotic learning datasets and rt-x models." arXiv preprint arXiv:2310.08864 (2023).
>
> [2] Kidambi, Rahul, et al. "Morel: Model-based offline reinforcement learning." Advances in neural information processing systems 33 (2020): 21810-21823.
>
> [3] Yu, Tianhe, et al. "Mopo: Model-based offline policy optimization." Advances in Neural Information Processing Systems 33 (2020): 14129-14142.
>
> [4] Haldar, Siddhant, et al. "Teach a Robot to FISH: Versatile Imitation from One Minute of Demonstrations." arXiv preprint arXiv:2303.01497 (2023).

---

### Official Review · Reviewer_bAbn · 2023-10-29

**Soundness:** 3 good
**Presentation:** 4 excellent
**Contribution:** 3 good
**Rating:** 8
**Confidence:** 3

**Summary:**

The learning process known as "sensory scaffolding" involves novice learners using more sensory inputs than experts. This principle has been applied in this study to train artificial agents. The researchers propose "Scaffolder," a reinforcement learning method that utilizes privileged information during training to optimize the agent's performance.

To evaluate this approach, the researchers developed a new "S3" suite of ten diverse simulated robotic tasks that require the use of privileged sensing. The results indicate that Scaffolder surpasses previous methods and frequently matches the performance of strategies with continuous access to the privileged sensors.

**Strengths:**

This paper delves into a critical question within the field of reinforcement learning: how can we effectively use privileged information as a 'scaffold' during training, while ensuring the target observation remains accessible during evaluation? This question takes on an added significance in robotic learning, where simulation is a major data source.

While there has been considerable research in this area, as detailed in the related work, this paper adds value to the existing body of knowledge, even without introducing novel methods. The proposed method may not be groundbreaking, but it offers a comprehensive examination of this issue from four perspectives: model, value, representation, and exploration.

This research serves as a valuable resource for those looking to deepen their understanding of the field. The excellent writing and presentation of this paper further enhance its contribution. Overall, despite the lack of methodological novelty, the paper is worthy of acceptance due to its systematic exploration and clear articulation of the subject matter.

**Weaknesses:**

1. Increasing the clarity around the Posterior and detailing how it is used to transition from the privileged latent state to the non-privileged latent state would greatly enhance understanding of the method.

2. The related work section could be expanded to include research papers that leverage privileged simulation reset to improve policy. These works also seem to align with the scaffolding concept presented in this paper. Papers such as [1][2] could be added for reference.

3. In the experimental design, the wrist camera and touch features don't appear to be excessively privileged or substantially different from the target observations. It would be beneficial to experiment with more oracle-like elements in the simulator as privileged sensory inputs. For instance, the oracle contact buffer or geodesic distance could be considered.


[1] DeepMimic: Example-Guided Deep Reinforcement Learning of Physics-Based Character Skills

[2] Sequential Dexterity: Chaining Dexterous Policies for Long-Horizon Manipulation

**Questions:**

1. More clarification on the posterior and embedding component in the method part.
2. More clarification of other scaffolding methods in the related work, no need for any experiments.

---

> ### Author Response · Authors · 2023-11-16
> **Author Response**
>
> Thank you for your review!
>
> > Increasing the clarity around the Posterior and detailing how it is used to transition from the privileged latent state to the non-privileged latent state would greatly enhance understanding of the method.
>
> We noticed several reviewers also had questions about the methodology, in particular the Nested Latent Imagination where we unroll the scaffolded dynamics model with the target policy and posterior. The reviewer may be interested in our response to Reviewer xL3j on the motivation behind learning separate scaffolded and target representations.
>
> We have added additional details in Appendix A.3 explaining the posterior and its usage in Nested Latent Imagination (NLI) to the appendix, along with pseudocode and a side-by-side figure comparing Nested Latent Imagination with the standard Latent Imagination procedure from DreamerV3 which only uses the impoverished target world model to generate trajectories.
>
> > Related work section could be expanded to include research papers that leverage privileged simulation reset to improve policy.
>
> That is an interesting point! While we mainly considered privileged training in the form of privileged sensors (hence the name sensory scaffolding), resets could be seen as another privileged training mechanism to scaffold policy learning. We have incorporated this discussion into the related work (changed text in red), citing the papers suggested.
>
> > In the experimental design, the wrist camera and touch features don't appear to be excessively privileged or substantially different from the target observations. It would be beneficial to experiment with more oracle-like elements in the simulator as privileged sensory inputs. For instance, the oracle contact buffer or geodesic distance could be considered.
>
> Scaffolder can be used for sim-to-real, with simulator states acting as the privileged sensors. In this scenario, it makes sense to use oracle-like elements in the simulator. Some tasks in our S3 suite do indeed resemble such a setting: for example, the dexterous manipulation tasks use ground truth object poses as privileged sensors.
>
> However, it’s important to note that Scaffolder generalizes privileged sensors from simulator state, to noisy, high-dimensional observations - enabling privileged training in the real world. Some of the S3 tasks are designed to emulate a real world learning setup and as a result use realistic privileged sensors such as wrist cameras or touch sensors.
>
> > While there has been considerable research in this area, as detailed in the related work, this paper adds value to the existing body of knowledge, even without introducing novel methods. The proposed method may not be groundbreaking, but it offers a comprehensive examination of this issue from four perspectives: model, value, representation, and exploration.
>
> While this may be subjective, we respectfully disagree with the assertion that this paper does not introduce novel methods. Scaffolder is a novel method that permits privileged information to influence and improve policy learning through a comprehensive suite of routes: world model, value, exploration, and representation, as acknowledged by  reviewers V3Vw and Sbzi. Firstly, it is not a priori clear how to achieve this, but furthermore, Scaffolder is more substantial than simply combining various existing methods. A particular salient example is Nested Latent Imagination, which frugally achieves the goals of generating synthetic experience for an impoverished target policy, from a world model powered by privileged sensory information. Note that others have also considered this problem and arrived at different solutions: see our baseline Informed Dreamer, which Scaffolder outperforms comprehensively.
>
> Finally, beyond the method, this paper is also novel in considering learning from privileged information beyond just simulator state: Scaffolder can use high dimensional and noisy privileged observations like RGB images to train target policies.

---

> > ### Comment · Reviewer_bAbn · 2023-11-22
> > **Replay to Author Rebuttal**
> >
> > Thanks for the detailed explanation. The idea of enable real world training with high-dimensional observations for scaffolding is fantastic. I really hope this paper can be accepted and so keep my original rating of strong accept.

---

### Official Review · Reviewer_V3Vw · 2023-11-01

**Soundness:** 3 good
**Presentation:** 3 good
**Contribution:** 3 good
**Rating:** 8
**Confidence:** 3

**Summary:**

This work proposes to utilize privileged sensory information to improve every component of model-based reinforcement learning, including world model, exploration policy, critic, and representation as well. This work provides extensive evaluation over 10 environments including different kinds of sensory data, showing the proposed method outperform all representative baselines. This work also provide detailed ablation study over all environments showing the

**Strengths:**

1. This work provides systematic analysis over different components in the “sensory scaffolding” setting, and proposes corresponding scaffolding counterparts  of every component in MBRL, except the policy during deployment.
2. This work provides a promising evaluation comparison with multiple representative baselines, demonstrating that with the proposed pipeline, privilege information improves the sample efficiency as well as the final performance over wide-range of tasks.
3. Through ablation study, this work shows different components in the system boost the performance in a different way, providing additional insight on how privileged information can be used in the future work.
4. Experiment details are well presented in the Appendix, including runtime and resource comparison over different methods on different environments.
5. The overall presentation of the work is good, considering the complexity of the system and amount of information delivered.

**Weaknesses:**

1. For scaffolded TD error comparison, it’s not clear why the comparison is conducted on Blind pick environment, since the gap between the proposed method and the version without scaffolded critic is much larger (at least in terms of relative gap) on Blind Cube Rotation environment. Also it would be great to see whether the estimate is close for tasks like Blind Locomotion (since the gap is small on that task). It seems there is some obvious pattern in the Figure 9, that the scaffolded TD is worse at 5, 10, 15 epoch and performs best on 7, 12, 18 epoch, it would be great to have some explanation for that.
2. For some claims made in the paper, it’s actually not quite convincing. For “In other words, much of the gap between the observations o− and o+ might lie not in whether they support the same behaviors, but in whether they support learning them.”, some additional visualization like trajectory visualization might be helpful to strengthen the claim, since the similar reward score does not necessarily result in similar behavior.
3. For runtime comparison, since the speed of given GPUs varies a lot, it might be better to compare the wall-time with similar system configuration, assuming the wall-time is consistent across different seeds.

**Questions:**

1. Refer to weakness.
2. Regarding some technical details, is a bit confusing:
* In section C1, it says “We launch 4-10 seeds for each method”, what’s the exact meaning of using different number seeds across methods or across environments?

---

> ### Author Response · Authors · 2023-11-16
> **Author Response Part 1 of 2**
>
> Thank you for your review!
> > For scaffolded TD error comparison, it’s not clear why the comparison is conducted on Blind pick environment, since the gap between the proposed method and the version without scaffolded critic is much larger (at least in terms of relative gap) on Blind Cube Rotation environment. Also it would be great to see whether the estimate is close for tasks like Blind Locomotion (since the gap is small on that task).
>
> Thank you, we believe this comment arises from a misunderstanding of the analysis experiment reported in Fig 9. Our exposition here grew rather terse for space, and we have updated the experiment description in Section 4.3 to improve it, following the clarifications below.
>
> Fig 9 *does not* represent a comparison between Scaffolder and No Scaffolder Critic. Instead, it compares the policy return estimated with all scaffolded components (specifically dynamics, reward, critic, and continue heads, as seen in Eq 3) with the policy return estimated using *no* scaffolded components. Recall that the gradient of this estimated policy return determines the policy updates.
>
> We ran this analysis on Blind Pick because, as Fig 8 shows, “No-Scaff. WM” performs very poorly here compared to full Scaffolder (this is also true for Blind Locomotion). We wanted to trace this performance gap all the way back to differences in the computed policy returns when not using scaffolded WM (including dynamics, reward, continue heads) and critic.
>
> We also agree with the reviewer that reporting these results for tasks other than Blind Pick may be interesting, and plan to report them, hopefully within the rebuttal period.
>
>
> > For some claims made in the paper, it’s actually not quite convincing. For “In other words, much of the gap between the observations o− and o+ might lie not in whether they support the same behaviors, but in whether they support learning them.”, some additional visualization like trajectory visualization might be helpful to strengthen the claim, since the similar reward score does not necessarily result in similar behavior.
>
> Thank you, we recognize now that this statement was not clearly worded. Target and privileged policies often behave very differently in our settings, of necessity: for example, consider Blind Pick, where the target policy must first grasp around the table to find an object (see [videos on the website](https://sites.google.com/view/sensory-scaffolding#h.qlmom6baw90)), whereas the privileged policy sees the object and can directly pick it up. We do not of course claim that such wildly different behaviors are the same, just that they might achieve similar task metrics (e.g. task success rates). We will reword to: “In other words, much of the gap between the observations o− and o+ might lie not in whether they support achieving similar task performance metrics, but in whether they support _learning_ them."
>
> > For runtime comparison, since the speed of given GPUs varies a lot, it might be better to compare the wall-time with similar system configuration, assuming the wall-time is consistent across different seeds.
>
> We will update the appendix with wall times recorded on a similar system configuration.
>
> > In section C1, it says “We launch 4-10 seeds for each method”, what’s the exact meaning of using different number seeds across methods or across environments?
>
> Due to computational limitations, it was expensive to run 10 seeds for all methods across all environments. We respectfully ask the reviewer to consider our large evaluation setup - 10 tasks and 5 model-based methods each taking a GPU and up to a day to run would result in 500 GPU days worth of compute.
>
> We have recorded the current number of seeds for each task below.
> |  | Blind Pick | Blind Locomotion | Blind Deaf Piano | Blind Numb Pen | Blind Numb Cube | Noisy Monkey | Wrist Pick-Place | Occluded Pick-Place | RGB Pen | RGB Cube |
> |---|---|---|---|---|---|---|---|---|---|---|
> | Scaffolder | 8 | 5 | 5 | 5 | 5 | 5 | 5 | 5 | 5 | 5 |
> | DreamerV3 | 10 | 10 | 5 | 5 | 5 | 5 | 5 | 5 | 4 | 4 |
> | Informed Dreamer | 8 | 5 | 5 | 5 | 5 | 5 | 5 | 5 | 5 | 5 |
> | DreamerV3 + BC | 5 | 5 | 5 | 5 | 5 | 5 | 5 | 5 | 4 | 4 |
> | Guided Observability | 5 | 5 | 5 | 5 | 5 | 5 | 5 | 5 | 5 | 5 |
> | RMA+ (100M) | 1 | 1 | 1 | 1 | 1 | 1 | 1 | 1 | 1 | 1 |
> | AAC (100M) | 1 | 1 | 1 | 1 | 1 | 1 | 1 | 1 | 1 | 1 |
>
> Note that our baselines RMA+ and AAC are run for 100M environment steps (20-400x the normal step budget), at which point we expect them to have overcome variations across initialization seeds. As a result, we just reported 1 seed for RMA+ and AMC.
>
> We are running more seeds, and will update the results with 10 seeds for all tasks when ready, although this will likely happen after the rebuttal period. We do not anticipate any major changes in trends.

---

> > ### Author Response · Authors · 2023-11-16
> > **Author Response Part 2 of 2**
> >
> > > It seems there is some obvious pattern in the Figure 9, that the scaffolded TD is worse at 5, 10, 15 epoch and performs best on 7, 12, 18 epoch, it would be great to have some explanation for that.
> >
> > We believe this is a spurious correlation, and did not observe this in other experiments. In an earlier experiment, we also ran the TD comparison on a blind grid world navigation task (see Appendix D). We did not observe any obvious patterns, and found the overall trends to still hold.

---

> > > ### Comment · Reviewer_V3Vw · 2023-11-22
> > > **Response**
> > >
> > > Thank you authors for clarification,
> > > I'll keep my original evaluation.

---

> ### Author Response · Authors · 2023-11-23
> **TD analysis on another environment for V3Vw (11/22/23)**
>
> We have now performed the TD($\lambda$) analysis experiment on another environment, Blind Locomotion, as you had suggested. We find similar trends where the Scaffolded TD($\lambda$) return estimate is more accurate than the target TD($\lambda$) estimate. This provides further evidence that the training signals from Scaffolder are more accurate and lead to better policy improvement. Note that this experiment requires training Scaffolder from scratch with additional components (namely the target critic) to permit computing the target TD($\lambda$) estimate in addition to the scaffolded TD($\lambda$) estimate, so we had reported it only for Blind Pick (Fig 9) in the main paper. We added this Blind Locomotion result to Appendix D as additional evidence, alongside the Gridworld results already reported there.
>
>
> We have provided some walltime learning curves, in response to xL3j, that you may also be interested in. Due to time constraints, we didn’t have time to rerun all methods on a single machine, but we will do so after the rebuttal.

---

### Official Review · Reviewer_xL3j · 2023-11-06

**Soundness:** 3 good
**Presentation:** 3 good
**Contribution:** 3 good
**Rating:** 8
**Confidence:** 2

**Summary:**

This paper proposes Scaffolder, a MBRL method that extends DreamerV3 with privileged information in its modules. Scaffolder uses privileged world models and exploration policies to roll-out trajectories to train a better target policy. To ensure consistency between target and privileged latent, Scaffolder proposes to predict target latent from privileged latent, bottlenecked by target observation. Scaffolder outperforms baselines on the newly proposed S3 benchmark.

**Strengths:**

+ The paper is well written and motivated. The presentation is clear.
+ Strong empirical performance.

**Weaknesses:**

- I agree that it makes sense to evaluate the proposed method on the newly proposed benchmark, for motivations mentioned in the paper. However, the paper would still benefit from evaluating extra existing benchmarks, just for reference.
- One major benefit of privileged information reinforcement learning is to train the target policy with privileged information in simulation, and deploy it in the real world where there is no privileged information. However, all experiments in the paper are purely in sim. Can the authors comment more on how well the presented approach will work in real-world applications?
- In addition to the number of frames being the x-axis for figure 6, please also include one where x-axis is the wall-clock time. This way the community will have a better understanding of how the proposed method and baselines work on this particular environment set.

**Questions:**

- Real-world applications. Please see details in the weaknesses section above.
- I am curious about one particular design choice. Why do the authors choose to predict target latent from privileged latent, bottlenecked by target observation. Why not usethe same latent, shared by both the privileged and target modules?

---

> ### Author Response · Authors · 2023-11-16
> **Author Response Part 1 of 2 (9/16/23)**
>
> Thank you for your review!
> > Sim-to-real scaffolder for real-world applications?
>
> Yes, Scaffolder can easily be used as a sim-to-real approach, where simulator state serves as the privileged observation and sensory inputs (e.g. camera images) can serve as the target observation. Among our experimental settings, RGB Cube and RGB Pen involving privileged object pose and target visual observations follows a similar template.
>
> Furthermore, note that we evaluate Scaffolder in more general settings that suggest a more direct route to real-world applications: even the privileged observations in our settings often come from additional sensors, rather than directly reading the simulator state. (e.g. privileged cameras in the Blind Pick environment) — this can be replicated in real-world learning settings such as the autonomous car setup described in the introduction. Scaffolder is also sample-efficient enough to reasonably permit this: it inherits and improves the sample-efficiency of Dreamer, which has already been applied to real-world robot learning [1].
>
> Finally, you may also be interested in our response to reviewer Sbzi, where we suggest some practical ways to fine-tune and use offline data for Scaffolder.
>
> [1] Wu, Philipp, et al. "Daydreamer: World models for physical robot learning." Conference on Robot Learning. PMLR, 2023.
>
> > Performance vs training wall-clock time?
>
> Appendix B in the submission already reports the total training time for Scaffolder and baselines, showing that Scaffolder trains in similar time to other baselines. The model-free baselines RMA and AAC both need much longer to train than the model-based approaches like Scaffolder. Scaffolder needs on average 30-40% more wall clock time per update step than base Dreamer v3, due to having to train an additional world model but generally makes up for this by training much more sample-efficiently. We will expand this section to provide plots like Fig 6 but with wall clock time on the x-axis.
>
> Finally, note that training time costs in most real-world applications would be dominated by the cost of collecting environment steps (“frames”), such as with a robot moving in the real-world at each step. As such, the number of environment steps is often the most relevant cost to consider, which is why research in this area usually reports performance vs. steps as in our Fig 6.
>
> > Design choice: why predict target latent from privileged latent, bottlenecked by target observation,  why not use the same latent, shared by both the privileged and target modules?
>
> Thank you for this question. We notice that several reviewers have had questions about this part of our approach, and will aim to expand and improve the exposition of Nested Latent Imagination, following the description below.
>
> Scaffolder tries to simultaneously achieve the following two things: (1) it trains a scaffolded world model operating on scaffolded observations $o^+$ (represented into a “scaffolded latent” $z^+$) to more faithfully model the environment and generate better synthetic training experiences, (2) it trains a target policy, which in our setting, is restricted to operate only on information from target observations $o^-$, represented in a target latent $z^-$. In other words, $z^-$ cannot contain information that cannot be inferred from target observations alone.
>
> A shared representation $z^+=z^-$, as used in some prior approaches like RMA and Informed Dreamer, would violate these requirements: if it contained information from privileged sensors to enable better world model learning as required in (1), it would not be fully inferable from $o^-$ alone, as required by (2). Consider, for example, S3 tasks like Blind Picking or Blind Locomotion, which have target observations (e.g. proprioception) that are not predictive of privileged observations (images of objects).
>
> Scaffolder proposes a relatively straightforward solution to achieve both (1) and (2). It keeps $z^+$ and $z^-$ separate, but learns a target embedding translator $z^+ \rightarrow e^-$. This allows a mapping from $z^+ \rightarrow e^- \rightarrow z^-$ via the target posterior, so that the synthetic experiences generated in the scaffolded WM can still be used to train target policies. Note that the scaffolded observations $o^+=[o^-, o^p]$ include and extend the target observations, so it is reasonable to learn a unique mapping from $z^+$ to $e^-$.

---

> > ### Author Response · Authors · 2023-11-16
> > **Author Response Part 2 of 2 (9/16/23)**
> >
> > > Extra results on existing benchmarks just for reference?
> >
> > Thank you for this suggestion. The majority of relevant prior works have been evaluated in a restricted setting where privileged sensors are always low-dimensional simulator states (e.g. AAC, RMA). In other cases, tasks were often much simpler: for example, consider the “Bumps” robotic task in [2], which is akin to a simpler version of the “Blind Pick” task in our S3 suite. In Bumps, a blind robot must feel around a tabletop to find randomly placed bumps using proprioception and touch, and the privileged observations reveal bump locations. We deliberately constructed S3 to have complex tasks such as dexterous manipulation, and high-dimensional / noisy privileged sensors like cameras rather than merely low-dimensional simulator states. As a result, S3 is, to our knowledge, the most general and most challenging suite of environments testing learning with privileged sensory information.
> >
> > However, we agree that a comparison on previously proposed tasks is valuable: we are setting up experiments in the Bumps environment described above, and will report on our progress before the rebuttal period closes. We expect broadly similar trends to Blind Pick since, as described above, these tasks have similar privileged and target sensor gaps.
> >
> > [2] Nguyen, Hai, et al. "Leveraging fully observable policies for learning under partial observability." Conference on Robot Learning. PMLR, 2022.

---

> > > ### Comment · Reviewer_xL3j · 2023-11-20
> > > **Thanks for the response**
> > >
> > > Thanks for addressing my questions. My concerns are resolved, and I am raising my scores accordingly.

---

> > > > ### Author Response · Authors · 2023-11-22
> > > > **Pre-existing benchmark Experiment Update to xL3j  (11/21/23)**
> > > >
> > > > Dear reviewer, you had suggested adding evaluations of Scaffolder on extra existing benchmarks for reference. In our earlier response to you (dtd. 11/16), we promised to investigate some environments from COSIL [1]. In our global response (dtd. 11/18), we reported that we had found the Bumps-1D environment to be trivially solved by our DreamerV3 baseline and therefore unsuitable for these experiments.
> > > >
> > > > We have now finished running Scaffolder and DreamerV3 (operating only on target input $o^-$, consistent with our other results) on two other environments from the COSIL paper, Bumps-2D and Car-Flag.
> > > >
> > > > In Bumps-2D, a robot arm must find the bigger bump out of a pair of randomly initialized bumps in a 2D grid, using only proprioception and touch sensing. In Car-Flag, a car using just proprioception must find a randomly initialized green flag. If the car finds a blue flag, the green flag position is revealed. In both environments, the state expert can directly go to the bigger bump / green flag, while the student policy must perform information gathering actions to solve the task.
> > > >
> > > > We have updated the website [with the plots](https://sites.google.com/view/sensory-scaffolding). The new results are consistent with those reported in our paper on the 10 S3 tasks: Scaffolder clearly outperforms DreamerV3, further establishing the versatility of our approach for utilizing privileged sensory information.
> > > >
> > > > In addition, these experiments also enable new comparisons with the prior work evaluated in the COSIL paper [1]. Note that COSIL and its baselines operate in a slightly different setting from Scaffolder: they assume access to a perfect, hand-coded scaffolded policy $\pi^+$, whereas Scaffolder trains both target and scaffolded policies from scratch, and in tandem, starting at environment step 0. This means that, when comparing learning curves, we must remember that COSIL effectively “starts ahead” of Scaffolder. *Even with this handicap, Scaffolder easily outperforms COSIL on Bumps-2D, and matches it on Car-Flag.* These results on COSIL’s own evaluation environments show further evidence of Scaffolder’s efficacy improvements over prior state of the art.
> > > >
> > > > [1] Nguyen, Hai, et al. "Leveraging fully observable policies for learning under partial observability." Conference on Robot Learning. PMLR, 2022.

---

> > > > ### Author Response · Authors · 2023-11-23
> > > > **Walltime Learning Curves for xL3j (11/22/23)**
> > > >
> > > > In addition to reporting total training wall clock time in Appendix B, we have now added the full task-wise learning curves (as in Fig 6) over hours instead of environment samples, as you had requested. As expected, the trends largely stay the same as the original curves plotted over samples - even in walltime, Scaffolder largely outperforms baselines across tasks.. Thus, even in cases where training computation time rather environment samples are the most expensive resource, Scaffolder still outperforms its baselines.

---

### Author Response · Authors · 2023-11-18
**Global Response (tl;dr of some common items & experiment update) (11/18/23)**

We thank the reviewers for their comments and efforts so far. We have individually responded to each reviewer on 11/16, and are running experiments where appropriate. In this global response post, we provide a tl;dr summary for some key items of interest to multiple reviewers, and provide an intermediate report of the main follow-up experiment that our reviewers have suggested.

- **Motivation and details of Scaffolder’s methodology (Reviewer xL3j, bAbn, Sbzi):** DreamerV3 uses the world model’s dynamics to generate data, and posterior to aggregate target observation history for policy execution. Scaffolder trains an additional scaffolded world model and uses the scaffolded dynamics to generate more accurate data. Only target observations are available during deployment time, necessitating the usage of the target WM’s posterior for policy execution.
- **Evaluating methods by wall-time instead of samples (Reviewer  xL3j, V3Vw):** Scaffolder is much more sample efficient than DreamerV3 despite being more computationally expensive, making up for slower wall times. Note that it is standard practice to evaluate by samples rather than walltime, as training times are oftentimes dominated by slow data collection (e.g. real robots or slow video games). We will add learning plots like Fig 6 but with wall clock time on the x-axis.
- **Applicability to other settings? (Reviewer xL3j, Sbzi):** We believe Scaffolder can be readily applied to real robot learning setups, either through sim-to-real or real RL. While we evaluated Scaffolder in online RL, we believe Scaffolder can be extended to offline / finetuning scenarios as well.

More details on each of these points are in the responses to individual reviewers posted on 11/16.

## Experiment update:
As we had promised to xL3j on 11/16, we are investigating the Bumps environments from [2]. Our initial experiments suggest that Bumps-1D is not well-suited to study sensor scaffolding methods because Dreamer with the impoverished target observation performs nearly identically and learns nearly as quickly to Dreamer trained and evaluated with privileged observations (see [plot](https://sites.google.com/view/sensory-scaffolding)), achieving similar performance to the method from [1].

This is consistent with our expectation that these environments might be too easy to get meaningful gains from training with privileged sensing. We are now investigating the 2D version of this task, still hoping to satisfy xL3j’s request for evaluation on some pre-existing environments.

[1] Nguyen, Hai, et al. "Leveraging fully observable policies for learning under partial observability." Conference on Robot Learning. PMLR, 2022.

---

> ### Author Response · Authors · 2023-11-23
> **All Experiments Completed (11/22/23)**
>
> We have now completed all three pending experiments requested by reviewers:
> - We evaluated Scaffolder's efficacy over SOTA baselines on a pre-existing privileged RL benchmark, as requested by Reviewer xL3j
> - We verified that Scaffolder yields more accurate learning signals on one more environment, Blind Locomotion, as requested by Reviewer V3Vw.
> - We plotted learning curves over walltime rather than environment samples and find that trends largely hold, as requested by Reviewer xL3j and V3Vw.
>
> The details of these experiments are reported in the individual responses to the reviewers. Overall, these new results add further support to the claims in the paper.
> We would like to express our sincere gratitude to all reviewers for the constructive feedback and for helping to improve our paper.

---

### Meta-Review · Area_Chair_bu1S · 2023-12-05

**Metareview:**

(a) The paper proposes "Scaffolder", a MBRL approach to train policies using privileged information available only at train time. The paper builds on DreamerV3, training privileged world models and privileged exploration policies that generate trajectory to train the target policy. The key insight is to use a "latent translator" to predict the latent state from the privileged latent state based on the target observation.
The paper introduces an evaluation suite named "S3," consisting of ten diverse simulated robotic tasks that require the use of privileged sensing. The results indicate that Scaffolder outperforms baselines and, in many cases, matches the performance of strategies with continuous access to privileged sensors.

(b) Strengths
(+) (xL3j, V3Vw, bAbn): The paper is well-written and clear, providing systematic analysis of the components in the sensory scaffolding setting, and demonstrating how privileged information improves sample efficiency and final performance across a wide range of tasks.
(+) (Sbzi): The paper introduces a novel evaluation suite, S3, for studying privileged MDPs. It demonstrates performance improvements against a number of competitive baselines, including Informed Dreamer.
(+) (bAbn): Accessible overview of related works and challenges in privileged RL.

(c) Weaknesses
(-)  (V3Vw) Insufficient theory to back claims: The paper makes a couple ambiguous claims on the impact of observation on the policies. It would benefit from connecting with the existing theory of decision-making in privileged information settings, e..g. the theory in [1]. Notably what are the conditions necessary for the learnt policy to have bounded performance against the optimal policy? The paper does not reference a long chain of papers that have struggled with this question (referenced in [1])

[1] Swamy, G., Choudhury, S., Bagnell, J., & Wu, S. Z. (2022). Sequence model imitation learning with unobserved contexts. Advances in Neural Information Processing Systems, 35, 17665-17676.

(-) (xL3j) Evaluation on existing benchmarks. While the authors include new benchmarks, providing results on existing benchmarks used in privileged RL tasks would allow for a more fair comparison
(-) (V3Vw) Poorly worded claim: The impact of observation on behavior learning is poorly worded and could use a more formally explanation and example.

PC Comment: After consideration, this paper has been upgraded to Spotlight

**Justification For Why Not Higher Score:**

While the paper is well-written and introduces an important benchmark, it doesn't do a good job characterizing where the method would work and where it would fail.

**Justification For Why Not Lower Score:**

The paper is well written, introduces an important benchmark and is of value to the community.

---

### Decision · Program_Chairs · 2024-01-16

Accept (spotlight)